# An in vitro human vessel model to study *Neisseria meningitidis* colonization and vascular damages

Léa Pinon[1], Melanie Chabaud[2], Pierre Nivoit[1], Jerome Wong Ng[2], Tri-Tho Nguyen[2], Vanessa Paul[1], Charlotte Bouquerel[2], Sylvie Goussard[1], Pauline Smilovici[1], Emmanuel Frachon[2], Dorian Obino[1], Samy Gobaa[2]*, Guilllaume Dumenil[1]*

[1]Institut Pasteur, Université Paris Cité, INSERM UMR1225, Pathogenesis of vascular infections, Paris, France; [2]Institut Pasteur, Université Paris Cité, Biomaterials and Microfluidics core facility, Paris, France

*For correspondence:
samy.gobaa@pasteur.fr (SG);
guillaume.dumenil@pasteur.fr
(GD)

Competing interest: The authors declare that no competing interests exist.

## eLife Assessment

The authors develop an **important** microfluidic microvascular model called "Vessel-on-Chip", which they use to study *Neisseria meningitidis* interactions within this in vitro vascular system. **Compelling** evidence shows that the fabricated channels are lined by endothelial cells, and these can be colonized by N. meningitidis that in turn triggers neutrophil recruitment. This model has advantages over the human skin xenograft mouse model, which requires complex surgical techniques, however, it also carries limitations in that only endothelial cells and supplied specific immune cells in the microfluidics are present, while true vasculature contains a number of other cell types including smooth muscle cells, pericytes, and components of the immune system.
[Editors' note: this paper was reviewed by Review Commons.]

**Abstract** Systemic infections leading to sepsis are life-threatening conditions that remain difficult to treat, and the limitations of current experimental models hamper the development of innovative therapies. Animal models are constrained by species-specific differences, while 2D cell culture systems fail to capture the complex pathophysiology of infection. To overcome these limitations, we developed a laser photoablation-generated, three-dimensional microfluidic model of meningococcal vascular colonization, a human-specific bacterium that causes sepsis and meningitis. Laser photoablation-generated hydrogel engineering allows the reproduction of vascular networks that are major infection target sites, and this model provides the relevant microenvironment reproducing the physiological endothelial integrity and permeability in vitro. By comparing with a human-skin xenograft mouse model, we show that the model system not only replicates in vivo key features of the infection, but also enables quantitative assessment with a higher spatiotemporal resolution of bacterial microcolony growth, endothelial cytoskeleton rearrangement, vascular E-selectin expression, and neutrophil response upon infection. Our device thus provides a robust solution bridging the gap between animal and 2D cellular models, paving the way for a better understanding of disease progression and developing innovative therapeutics.

## Introduction

Infectious diseases remain a significant global health burden, underscoring the need for continued advances in diagnosis, prevention, and treatment strategies. Developing novel therapeutic strategies against infections requires a deep understanding of the underlying physiopathological mechanisms

and robust experimental models that capture their hallmark features. The lack of models replicating the 3D infection environment while enabling quantitative assessments is slowing the pace of infection research. Overcoming this limitation is thus essential to drive progress in infectious disease research and address the growing challenge of antimicrobial resistance (*Murray et al., 2022*). In this study, we focused on meningococcal disease, which is a model system for systemic infectious diseases that are particularly severe forms of infection. Furthermore, as for most pathogens, meningococcal disease is highly human-specific, thus raising specific challenges for the development of experimental models.

Clinical studies have provided the key elements of *N. meningitidis* pathogenesis leading to the vascular damages observed during *purpura fulminans*. Histological studies of postmortem samples have shown that bacteria are found primarily inside blood vessels of different organs, including the liver, the brain, the kidneys, and the skin (*Mairey et al., 2006*). Bacteria are typically found associated with the endothelium on the luminal side in tight aggregates (*Melican and Dumenil, 2012*). The infection is associated with signs of vascular function perturbations, including congestion, intravascular coagulation, and loss of vascular integrity (*Faust et al., 2000*). The presence of bacteria within the lumen of blood vessels is also associated with an inflammatory infiltrate, mainly composed of neutrophils and monocytes (*Guarner et al., 2004*). A valid model of meningococcal infection should thus reproduce the interaction of bacteria with the endothelium and immune cells, as well as infection-induced vascular damage.

To overcome the human-species specificity of meningococcal infections in an animal model, a human skin xenograft mouse model was developed (*Melican et al., 2013*). This model was instrumental in demonstrating the importance of bacterial type IV pili (T4P) for vascular colonization (*Melican et al., 2013*) and neutrophil recruitment in infected venules (*Manriquez et al., 2021*). The neutrophils interacting with the infected human endothelium are of murine origin, adding complexity to the interpretation of these interactions and the resulting immune response. In addition, while providing the proper tissue context for meningococcal infection, this model is dependent on access to human skin, complex surgical procedures, and animal use.

Another popular experimental approach involves the use of endothelial cells in in vitro culture, where interaction between *N. meningitidis* and single human cells are studied on flat 2D surfaces (*Doulet et al., 2006*; *Bonazzi et al., 2018*; *Denis et al., 2019*; *Alonso-Roman et al., 2024*). Such models have been used to decipher the molecular interplay underlying bacteria-bacteria and bacteria-endothelial cell interactions and have been essential in characterizing the host cell reorganization upon bacterial adhesion (*Merz and So, 1997*; *Charles-Orszag et al., 2018*). This notably includes the remodeling of the plasma membrane with the formation of filopodia-shaped protrusions that stabilize the microcolony in the presence of flow-induced shear stress (*Mikaty et al., 2009*). The actin cytoskeleton has also been shown to be highly reorganized underneath bacterial microcolonies, forming a structure reminiscent of a honeycomb, also called cortical plaque (*Merz and So, 1997*; *Soyer et al., 2014*). Although easy to use, 2D models do not recapitulate the geometry of the vascular system, the proper molecular signaling of inflammation, or the vesicular transport of proteins, all found in a 3D microenvironment (*Offeddu et al., 2019*; *Wang et al., 2020*). An in vitro model of meningococcal infections that recapitulates these features in a biologically relevant 3D environment is still lacking.

Producing synthetic yet functional human vasculature has been a dynamic field of research over the last decade, including establishing a 3D vascular system inside a hydrogel loaded onto a microfluidic chip. Endothelial cells are embedded into hydrogels (e.g. fibrin) to self-assemble into interconnected tubes (*Kim et al., 2013*; *Ko et al., 2019*; *Haase et al., 2020*; *O'Connor et al., 2022*). Although delivering functional vasculature, this method does not allow for control of the produced vascular geometry. Other methods consist of loading cells into pre-formed structures using molding (*Haase and Kamm, 2017*; *Dessalles et al., 2021*), or viscous fingering (*Herland et al., 2016*). These approaches allow the formation of large patterns (120-150 μm) and straight structures, but vessels in tissues are tortuous, ramified, and can be as narrow as 10 μm in diameter. An alternative way for the production of synthetic vasculatures takes advantage of the photoablation technique (*Sarig-Nadir et al., 2009*; *Brandenberg and Lutolf, 2016*; *Enrico et al., 2022*). This technique comes with several advantages, including compatibility with a wide range of hydrogels and maximal control over the produced geometry.

In this study, we used a homemade laser ablation setup to engineer a vascular system on-chip with tunable parameters to generate vessels of various shapes and sizes, hence reflecting the complexity

of the in vivo vascular system. We were able to reproduce *Neisseria meningitidis* vascular colonization in vitro. Our Vessel-on-Chip model allowed simulation of the in vivo environment during the onset of meningococcal infection with higher spatiotemporal resolution using multidimensional time-lapse imaging while respecting the human specificity of the meningococcal infection. It replicates key features of the infection, including bacterial adhesion, cellular remodeling, vascular damage, and neutrophil response in infected vessels. The human skin xenograft animal model of meningococcal intravascular infection served as the gold standard to validate our Vessel-on-Chip system. This platform offers a robust bridge to animal and 2D models, providing a physiologically relevant and quantitative 3D tool for studying host-pathogen interactions in vitro.

## Results

### Replicating the geometries of infected vessels using photoablation

*N. meningitidis* infects various types of blood vessels in human cases and in the skin xenograft mouse model (*Figure 1—figure supplement 1A*), including arterioles, venules, and capillaries, which typically range from 10-100 µm in diameter, with an average size of 40-60 µm (*Manriquez et al., 2021*). Furthermore, infected vessels are typically branched and exhibit diverse geometries (*Figure 1A*). Thus, to study *N. meningitidis* vascular colonization and subsequent vascular damage in relevant vascular geometries, we developed a Vessel-on-Chip (VoC) model using photoablation to allow for complex geometries (*Brandenberg and Lutolf, 2016*). The chip features a central channel connected to two larger lateral channels and filled with collagen I-type gel, which acts as an extracellular matrix (*Herland et al., 2016*). The photoablation process involves the 3D carving of the bulk collagen I matrix at the center of the microfluidic chip with a focused UV laser beam (*Figure 1B*), which was tuned to control vessel dimensions (*Figure 1—figure supplement 1B*). Briefly, the chosen design is digitized into a list of positions to ablate. A pulsed UV-laser beam is injected into the microscope and shaped to cover the back aperture of the objective. The laser is then focused on each position that needs ablation. After introducing endothelial cells (HUVEC) in the carved regions, they formed a vascular lumen by adhering to the substrate (*Figure 1C*) and deformed the initially squared structure, leading to a circular architecture (*Figure 1—figure supplement 1C*). The lateral channels remain open to ensure regular liquid flows for nutrient replenishment and the later introduction of bacteria and immune cells.

In vivo, the extracellular matrix is crucial for providing structural and organizational stability (*Haase et al., 2022*; *Davis and Senger, 2005*), thus warranting optimization in our VoC. Endothelial cells intensively sprouted at collagen concentrations of 2.4 mg/ml (*Figure 1—figure supplement 1D, E*), a phenomenon we sought to suppress to better align with the physiological conditions in vivo (*Dudley and Griffioen, 2023*). Sprouting decreased with increasing collagen concentrations (*Figure 1—figure supplement 1E, F*), which correlates with the increase in gel stiffness (*Figure 1—figure supplement 1G*). However, while two different collagen gels (3.5 mg/ml Corning and 2.4 mg/ml Fujifilm) exhibited a comparable elastic modulus (50 Pa), they showed different effects on the number of endothelial cell sprouts. These results suggest that, in our model, endothelial cell sprouting is determined by both the concentration and composition of collagen, which likely vary in terms of cross-linkage and pore size. Based on these findings, we used a minimum of 4 mg/ml collagen gels for the rest of our study.

The introduction of endothelial cells led to the formation of straight hollow tubes with a circular cross-section (*Figure 1D*). The system also allowed the replication of more complex 3D structures (*Figure 1E*) and geometries of in vivo blood vessels, based on intravital imaging data. For instance, a vascular branching point, observed in a human vessel imaged in the human skin xenograft mouse model (*Figure 1A*), was accurately replicated in the VoC (*Figure 1F*). The resulting in vivo-like structure in the VoC preserved the aspect ratio of the original vascular geometry and could be further visualized at higher microscopy resolutions compared to blood vessels in animal models. In summary, the photoablation technique, when combined with optimized extracellular matrix properties, enables controlling the construction of reproducible on-chip vessels that closely mimic those targeted by the meningococci in vivo.

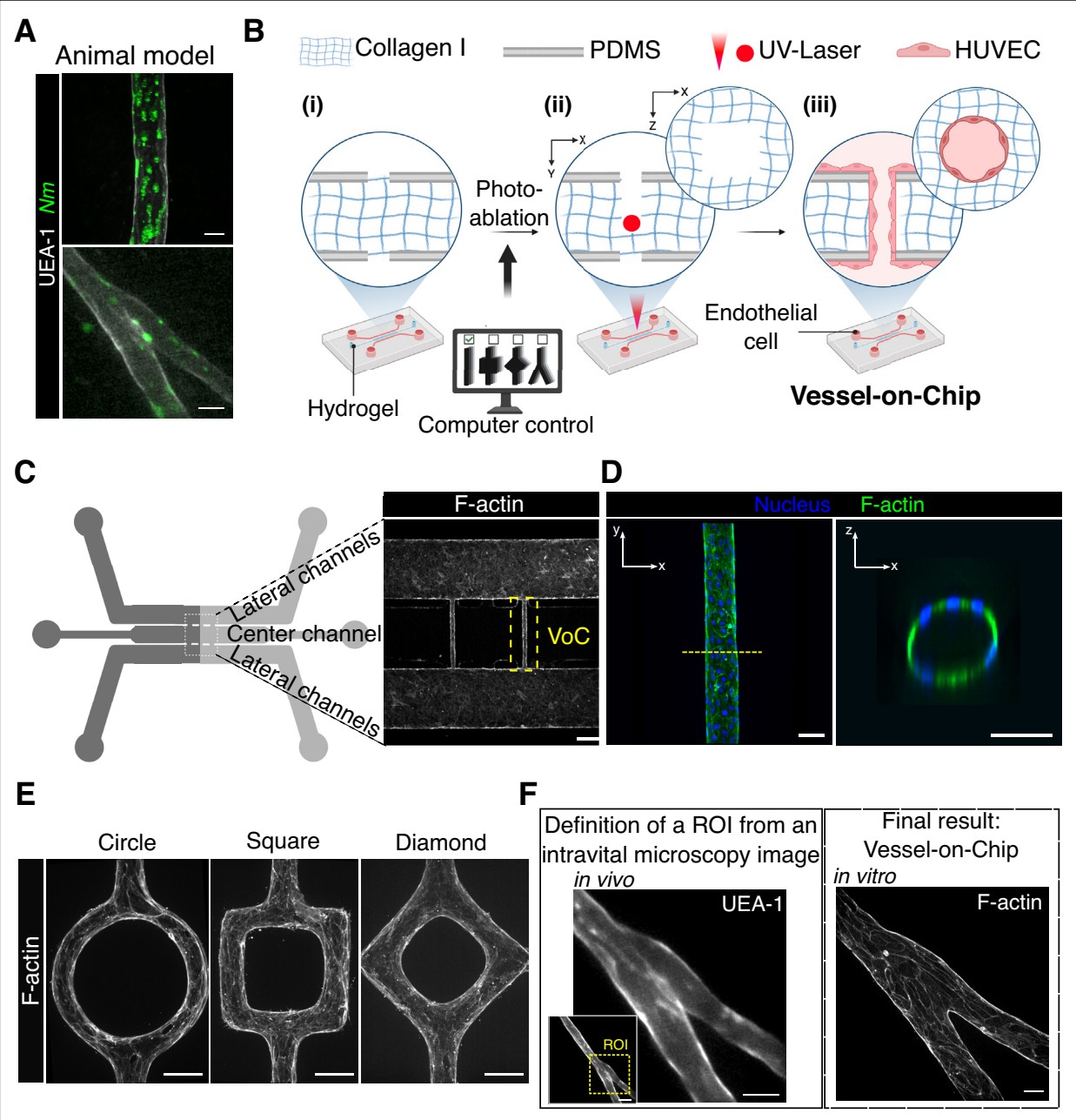

**Figure 1.** Replicating the geometries of infected in vivo vessels using photoablation. (**A**) Confocal images of *N. meningitidis*-infected vessels in the human-skin xenograft mouse model. Scale bar: 25 μm. (**B**) Schematic representation of the development of the Vessel-on-Chip (VoC) device: (i) a collagen-based hydrogel is loaded in the center channel of the microfluidic device, (ii) the focused UV-laser locally carves the chosen geometry within the collagen I matrix, (iii) HUVECs are seeded and attach on the collagen-carved scaffold. (**C**) Schematic representation of the microfluidic device and zoom of the carved region after cell seeding (F-actin). Scale bar: 250 μm. (**D**) Confocal images of F-actin and the related orthogonal view of the tissue-engineering VoC. Scale bar: 50 μm. (**E**) Confocal images of VoC in circle-, square-, and diamond-shaped structures. Scale bar: 50 μm. (**F**) Photoablation process to create in vivo-like structures. Definition of the region of interest (ROI) to replicate from intravital microscopy images (left), and the obtained in vivo-like structures in the VoC (right). Scale bars: 20 μm.

The online version of this article includes the following figure supplement(s) for figure 1:

**Figure supplement 1.** Development, optimization, and characterization of a tissue-engineered Vessel-on-Chip platform based on photoablation.

## The vessels in the VoC exhibit permeability levels comparable to those observed in vivo

A key feature of meningococcal infections is their association with an increase in vascular permeability (*Faust et al., 2000*; *Manriquez et al., 2021*). Therefore, relevant in vitro infection models should allow the observation and quantification of such a loss of vascular integrity. In this study, we assessed vascular

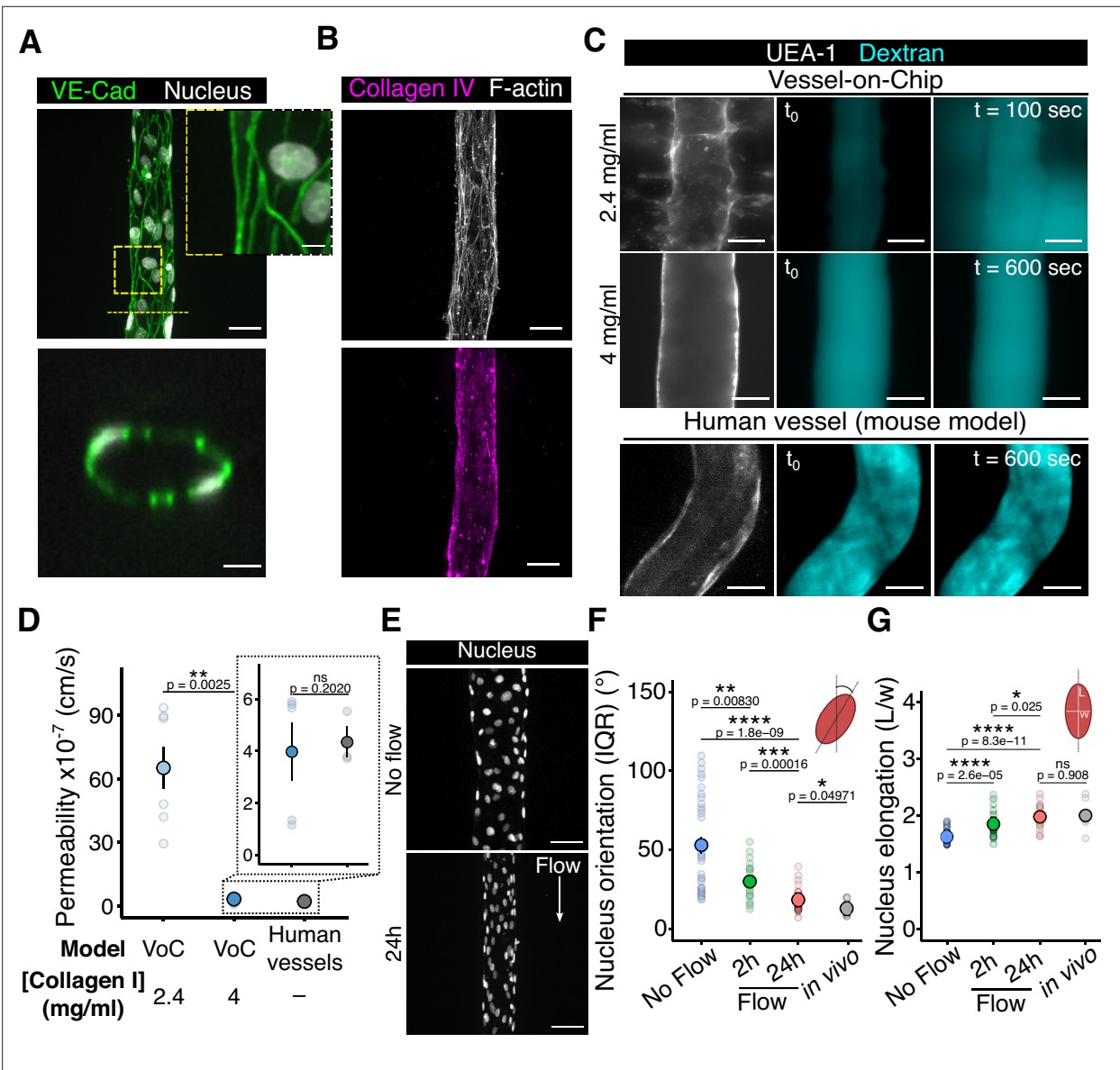

**Figure 2.** The Vessel-on-Chip device provides nuclear morphologies under flow conditions and permeability levels similar to those observed in vivo. (**A**) Confocal images of the VE-Cadherin staining and (**B**) Collagen IV in the VoC. Scale bars: 50 μm (large view) and 7 μm (zoom). (**C**) Representative confocal images of fluorescent 150 kDa-Dextran (FITC) in the VoC (top, scale bar: 50 μm) and in the human vessel in the mouse model (bottom, scale bar: 30 μm). Fluorescence of the outside and inside regions of the vascular lumen has been measured to determine permeability. (**D**) Graph representing the permeability to 150 kDa-Dextran. Each dot represents one vessel. For each condition, the mean ± s.d. is represented (VoC, 2.4 mg/ml: 65.12 ± 26.00 cm/s (n=7) – VoC, 4 mg/ml: 3.97 ± 2.49 cm/s (n=5) - Human vessel in vivo: 8.00 ± 2.52 cm/s (n=7)). (**E**) Representative images of nucleus alignment in the absence and presence of flow (24 hr). Scale bar: 50 μm. (**F–G**) Graphs of nucleus orientation (IRQ: interquartile range) and elongation. Each dot corresponds to the mean value of one vessel. For each condition, the mean ± s.d. is represented (No flow: 53.0 ± 30.2° and 1.63 ± 0.11 a.u. (n=34)–2 hr of flow: 29.9 ± 11.5° and 1.85 ± 0.25 a.u. (n=22)–24 hr of flow: 18.2 ± 7.61° and 1.98 ± 0.20 a.u. (n=25) – in vivo: 12.9 ± 4.6° and 2.01 ± 0.23 a.u. (n=9)). All statistics have been computed with Wilcoxon tests.

The online version of this article includes the following figure supplement(s) for figure 2:

**Figure supplement 1.** Endothelial cells in the Vessel-on-Chip devices respond to flow-induced shear stress.

permeability in both the VoC and in vivo blood vessels of human (skin xenograft mouse model) origin. Given that vascular integrity and permeability are dependent on intercellular junctions, we first visualized these structures in the chips. Two days post-seeding, endothelial cells formed thin and uniform junctions, as evidenced by VE-Cadherin (*Figure 2A*) and PECAM-1 (*Figure 2—figure supplement 1A*) stainings, confirming the cohesiveness of the engineered vessel. Additionally, we observed that endothelial cells secreted collagen IV (*Figure 2B*), a key component of the basement membrane that provides structural support to blood vessels (*Leclech et al., 2020*). Then, to evaluate vascular integrity functionally, we performed imaging-based permeability assays using fluorescent 150 kDa dextran (*Figure 2C*), comparing fluorescence intensities inside and outside the engineered vessels within our VoC. We observed that sprouting vessels (2.4 mg/ml collagen I) exhibited high permeability, whereas those without sprouts (4 mg/ml collagen I) showed low permeability – no leakage was detected for up to 10 min, consistent with in vivo observations in the human skin xenograft (*Figure 2D*). Altogether, these results demonstrate that our VoC forms a stable physical barrier with low permeability, closely mimicking in vivo conditions of human blood vessels in the animal model.

Then, as flow-induced shear stress is a key factor that shapes endothelial cell physiology (*Zhou et al., 2023a*), as well as *N. meningitidis* adhesion along the endothelium (*Mairey et al., 2006*), we ensured that our model recapitulated in vivo nuclear morphologies, known as sensors of the blood flow direction (*Davies, 1995*). By performing particle image velocimetry (PIV) experiments in anesthetized animals (*Figure 2—figure supplement 1B*), we measured the flow rates in blood vessels (*Figure 2—figure supplement 1C*) and computed the related wall shear stress (*Figure 2—figure supplement 1D*). Considering blood as Newtonian between 25 μm and 100 μm (*Pries et al., 1995*), we showed a wall shear stress of 1.57 Pa (15.7 dynes/cm$^2$) and 0.31 Pa (3.1 dynes/cm$^2$) in arterioles and venules, respectively – consistent with previous studies (*Davies, 1995*; *Osinnski et al., 1995*; *Peng et al., 2019*). Under venule-like flow conditions controlled by a syringe pump (*Figure 2—figure supplement 1E*), the nucleus of endothelial cells aligned along the flow direction (*Figure 2E*) and reached both elongation and orientation of in vivo features after 24 hr of flow exposure (*Figure 2F, G* and *Figure 2—figure supplement 1F*). Of note, nucleus orientation and elongation reached a constant value at 24 hr (*Figure 2—figure supplement 1G, H*), suggesting a homogeneous behavior regardless of wall shear stress for long-term flow exposure. Also, vessel diameter decreased when endothelial cells were subjected to flow, while cell density remained constant across all conditions (*Figure 2—figure supplement 1I, J*). Altogether, these results show that our VoC model correctly responds to flow-induced shear stress and recapitulates the physiological features of the animal model in terms of vascular nucleus morphology and permeability levels.

## Adhesion of *N. meningitidis* on chip depends on type-4 pili, while geometry-induced shear stress variations have little impact

The developed in vitro vessels allowed us to study meningococcal infection in a controlled 3D environment. Bacteria were introduced into the microfluidic device and circulated through the lumen (*Figure 3A*). Consistent with observations in infected patients and in the human skin xenograft mouse model (*Melican et al., 2013*), meningococci were found to attach to the vascular wall within the infected VoC (*Figure 3B*). *N. meningitidis* covered the three-dimensional areas of the endothelium and adhesion occurred both in straight and in vivo-like structures (*Figure 3B*), forming 3D microcolonies of varying sizes, ranging from a few to tens of microns, as observed in the original geometry in vivo. Overall, the infected Vessel-on-Chip model replicates meningococcal vascular colonization in a 3D context and enables high-resolution fluorescence microscopy for detailed monitoring of the infection process in vitro.

We confirmed the role of type IV pili-mediated adhesion in the VoC model by comparing a wild-type strain (WT) with the type IV pili-deficient *pilD* mutant strain. After 3 hr of infection under flow, we observed that infection of the VoC with the *pilD* mutant strain only contained rare *N. meningitidis* aggregates (*Figure 3C*). In contrast, infection with the WT strain resulted in the formation of numerous and large microcolonies on the vascular walls of the VoC (*Figure 3D*, *Figure 3—figure supplement 1A and B*).

We next investigated whether vascular geometry and associated shear stress variations affect *N. meningitidis* adhesion, i.e., whether *Neisseria meningitidis* exhibits preferential sites of infection across vascular geometries. We hypothesized that, if bacteria preferentially adhered to specific

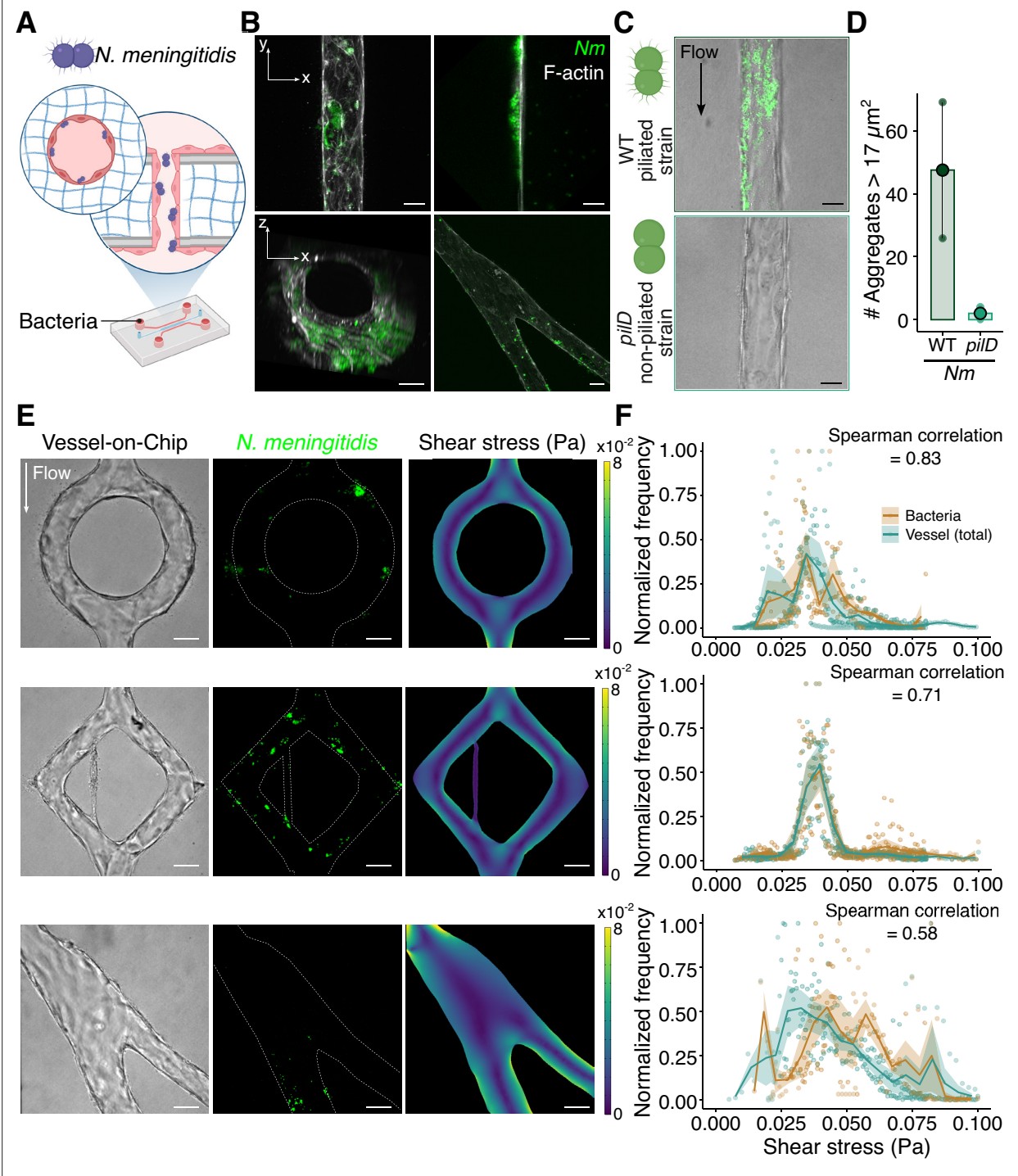

**Figure 3.** *Neisseria meningitidis* adhesion to the Vessel-on-Chip device. (**A**) Schematic of the infected Vessel-on-Chip device. (**B**) Confocal images of infected vessels in the VoC system 3 hr post-infection, in each condition. Top-left: z-projection of the infected VoC; top-right: slice of the middle of the infected VoC; bottom-left: 3D reconstruction of the infected VoC; and bottom-right: z-projection of the infected in vivo-like VoC. Scale bar: 20 µm. (**C**) Bright-field and fluorescence confocal images of the VoC 3 hr post-infection with WT (top) and *pilD* (bottom) *Nm* strains. Scale bar: 25 µm. (**D**) Graph representing the number of aggregates 3 hr post-infection, with wild-type (WT) and *pilD Nm* strains. 17 µm² represents the median over the entire population of aggregate sizes. Each dot corresponds to a vessel. For each condition, the mean ± s.d. is represented (WT: 47.5 ± 30.4 (n=2) – *pilD*: 2.0 ±2.0 (n=3)). (**E**) Brightfield (left) and fluorescence (center) images of the infected vessels of different designs (circle, diamond, in vivo-like) and related simulation of shear stress. Scale bar: 50 µm. (**F**) Normalized histograms of the total number of pixels (blue dots and curve) and bacteria pixels (orange

*Figure 3 continued on next page*

*Figure 3 continued*

dots and curve) depending on the shear stress values for each vessel design (from top to bottom, circle, diamond, in vivo-like). Solid curves represent the mean ± sd (n=3 experiments). Spearman correlation gives the correlation between the mean curves.

The online version of this article includes the following figure supplement(s) for figure 3:

**Figure supplement 1.** *Neisseria meningitidis* uses T4P to adhere on the Vessel-on-Chip and its adhesion does not depend on the difference of shear stress.

regions, the local shear stress at these sites would differ from the overall distribution. To this end, we infected vessel-on-chip models of increasing geometrical complexity, circle, diamond, and in vivo-like branched structures, generating distinct wall shear stress profiles (*Figure 3E*), from 0.025 to 0.1 Pa (*Figure 3F*). Thanks to a segmentation routine coupled with Comsol simulations (*Figure 3—figure supplement 1C*), we compared the shear stress at bacterial adhesion sites in the VoC (orange dots and curve) with the shear stress along the entire vascular edges (blue dots and curve) (*Figure 3F*). The distribution of adherent bacteria as a function of shear stress level (orange curve) closely mirrored the shear stress distribution available in the device (blue curve), indicating no preferential adhesion sites across shear stress gradients. This was confirmed by Spearman correlation analysis, which showed strong correlations between the two mean curves (0.58 < S. corr. < 0.83) (*Figure 3F*). These findings suggest that within this flow range, adhesion is not impacted by geometry-induced shear stress variations, consistent with in vivo observations and previous studies (*Mairey et al., 2006*).

## Progression of *N. meningitidis* colonization in the Vessel-on-Chip is similar to in vivo observations

In both the VoC and animal model, we observed the fusion of bacterial aggregates (*Figure 4—figure supplement 1A*), highlighting the dynamical properties of bacterial expansion (*Bonazzi et al., 2018*; *Rossy et al., 2023*). We then challenged the impact of flows in bacterial growth and morphology. Microcolony expansion was thus assessed under flow (0.5 µl/min which corresponds to ~ 0.1 Pa) by monitoring the increase of surface area covered by adherent bacteria over 3 hr of infection, as in vivo experiments (*Figure 4A* and *Figure 4—figure supplement 1B*). Individual microcolonies (shaded curves, *Figure 4A*) exhibited a mean linear growth trend (solid curve) in both the VoC model and the mouse model. We normalized the quantification of the colony doubling time according to the first time point where a single bacterium is attached to the vessel wall. We observed that the time required for microcolony expansion in both VoC (with and without flow) and human skin-grafted mice is similar (approximately 20 min, *Figure 4B*), and is lower than in agar-pad cultures (30-40 min *Ershov et al., 2022*). This trend likely reflects a combination of bacterial proliferation, fusion, and subsequent adhesion of circulating bacteria, as previously proposed (*Eugène et al., 2002*). We then observed that, under flow conditions, bacterial microcolonies were submitted to a morphological change within the VoC (*Figure 4C and D*). Microcolonies elongated and aligned along the vessel length, corresponding to the direction of flow. Unlike under static conditions, both elongation and orientation properties of bacterial microcolonies in the presence of flow were similar to those found in vivo, demonstrating the importance of the flow to recapitulate the proper infection conditions in the VoC. Altogether, these findings demonstrate that our model effectively replicates the key aspects of *N. meningitidis* colonization, including bacterial dynamics, morphology, and functional behavior, closely mirroring the infection process observed in vivo.

## *N. meningitidis* vascular colonization increases vessel permeability within the Vessel-on-Chip

Vascular permeability was shown to increase at the late stages of meningococcal infections (24 hr post-infection), as evidenced in the human skin xenograft mouse model by the accumulation of intravenously injected Evans blue in the surrounding tissues (*Manriquez et al., 2021*). We, therefore, used a direct permeability assay using a 150 kDa FITC Dextran in both the VoC and the mouse model (*Figure 4E and F*) to determine the permeability at the early stage of inflammation and infected vessels. Histamine was used as a positive control (*Egawa et al., 2013*), showing a similar response under inflammatory conditions in both the in vivo and VoC models (*Figure 4E and F*). After 3 hr post-infection, vascular permeability importantly increased in the infected VoC, consistent with findings

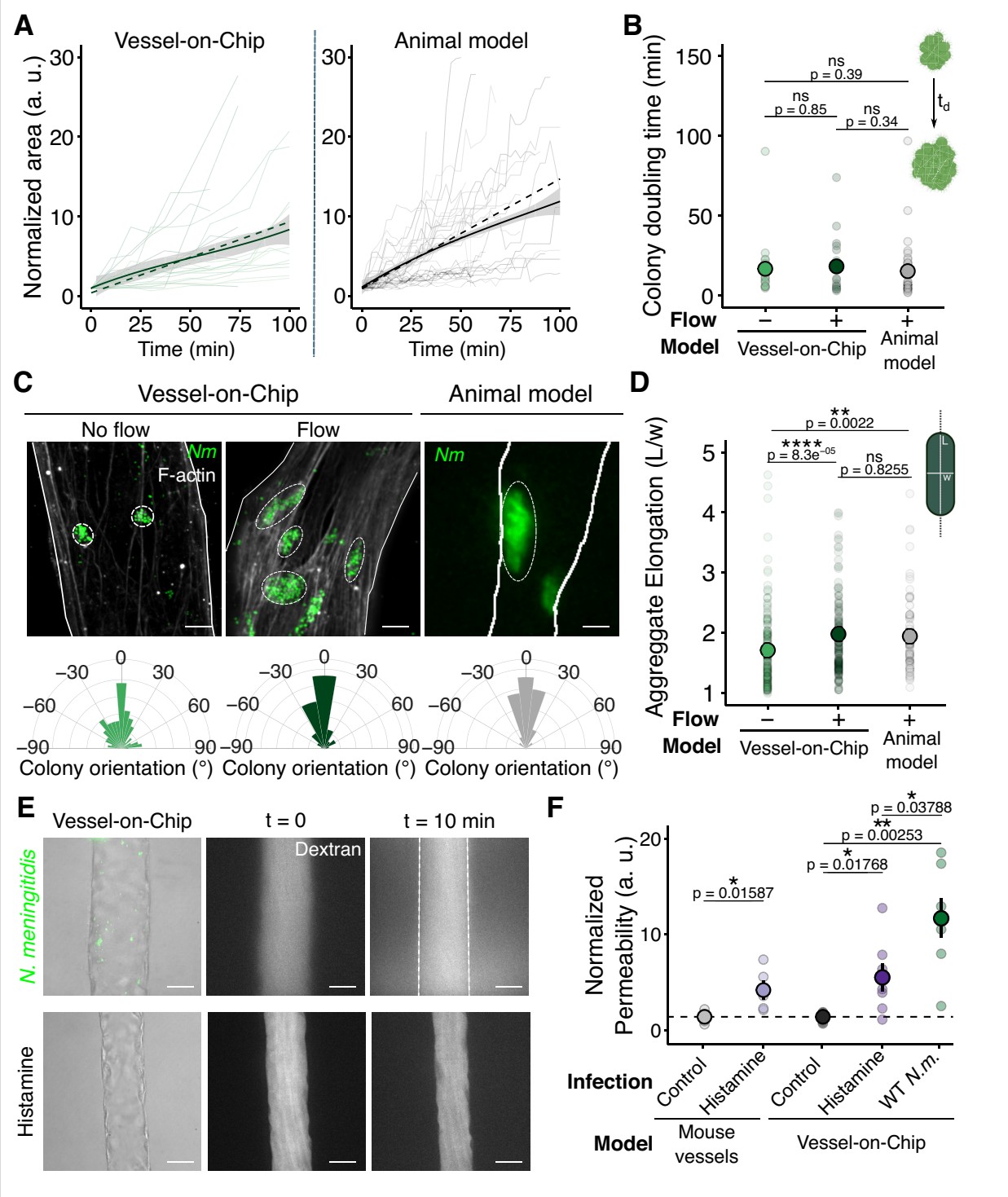

**Figure 4.** Flow impacts colony morphology but not growth in both Vessel-on-Chip and animal models, and *N. meningitidis* infection increases permeability in Vessel-on-Chip. (**A**) Normalized surface area of microcolonies over time, in the VoC under flow conditions (left, n=20) and in human vessels of the xenografted mouse model (right, n=33). Solid thin curves, solid thick curves, and dashed thick curves represent individual colony growth, mean, and linear fit (y=ax + 1), respectively. (**B**) Colony doubling time extracted from the curves ($t_d$ = 1/a). Each dot corresponds to a bacterial microcolony. For each condition, the mean ± s.d. is represented (VoC, Flow⁻: 16.7 ± 18.5 min (n=21) — VoC, Flow⁺ : 26.2 ± 40.4 min (n=20) — in vivo: 15.2 ± 18.8 min (n=33)). (**C**) Confocal images of wild-type (WT) *Nm* microcolonies formed on the vascular wall 3 hr post-infection, in the absence and presence of flow (VoC), and in the animal model. Scale bar: 10 μm. Circular plots representing the distribution of microcolony orientation. (**D**) Graph representing the elongation of microcolonies 3 hr post-infection in each condition. Each dot corresponds to a microcolony. For each condition, the

*Figure 4 continued on next page*

*Figure 4 continued*

mean ± s.d. is represented (VoC, Flow⁻ : 1.75 ± 0.77 (n=128) — VoC, Flow⁺ : 2.11 ± 1.02 (n=125) — in vivo: 2.07 ± 0.971 (n=61)). (**E**) Brightfield and fluorescence confocal images of permeability assay (Dextran) in infected (top) and histamine-treated (bottom) Vessel-on-Chips. Scale bar: 25 μm. (**F**) Relative permeability values of VoC and in vivo vessels treated with histamine and infected with *Nm*. Each dot represents a vessel. For each condition, the mean ± s.d. is represented (mouse vessel, Control: 1 ± 0.62 (n=5), Histamine: 3.79 ± 2.25 (n=5) — VoC, Control: 1 ± 0.63 (n=5), Histamine: 5.11 ± 3.92 (n=7), *Nm*: 11.3 ± 5.48 (n=7)). All statistics have been computed with Wilcoxon tests.

The online version of this article includes the following figure supplement(s) for figure 4:

**Figure supplement 1.** *N. meningitidis* colonies grow and reorganize in vitro under flow conditions, and in vivo permeability is not affected by *N. meningitidis* at early stages of infection.

from previous 2D in vitro studies (**Kobsar et al., 2011**; **Endres et al., 2022**; **Schubert-Unkmeir et al., 2010**; **Ziveri et al., 2024**). In contrast, only a subtle increase in vascular permeability was observed in infected human vessels in the skin xenograft mouse model (**Figure 3—figure supplement 1C**), most likely reflecting the presence of additional cell types surrounding the vessel wall in vivo, which enhance barrier integrity at early stages of infection (**Endres et al., 2022**). In addition, at later stages of infections, vascular integrity might also be affected by intravascular coagulation induced by meningococci (**Corre et al., 2022**). Overall, our results show that, despite moderate differences between the VoC and the animal model in the kinetics of endothelial integrity loss, the VoC offers a suitable model to study the direct consequences, including the increase in vessel permeability, of the host-pathogen interaction between *N. meningitidis* and endothelial cells upon vascular colonization with a high spatiotemporal resolution.

## Flow-induced aligned actin stress fibers are reorganized beneath bacterial microcolonies

*N. meningitidis* bacterial microcolonies have been shown to reorganize both the plasma membrane and the actin cytoskeleton upon adhesion at the surface of endothelial cells in 2D models (**Charles-Orszag et al., 2018**; **Soyer et al., 2014**; **Merz et al., 1999**). However, in vivo, such events are challenging to observe.

Also, endothelial cells experience flow-induced mechanical stresses in vivo, which is known to influence the state of the actin cytoskeleton (**Malek and Izumo, 1996**; **Davies, 2009**). The impact of such actin cytoskeleton rearrangements, potentially impacting bacteria-induced responses, has never been explored. This requires a high-resolution imaging of infection within a perfused 3D environment, which is provided by our system. Thus, within the micro-physiological environment of the Vessel-on-Chip model, we investigated whether similar actin reorganization in endothelial cells occurs upon infection by *N. meningitidis*. We first confirmed that, in the absence of flow, actin stress fibers exhibit random orientations (**Figure 5A and B**). In contrast, when endothelial cells in the VoC were submitted to flow for 2 hr, actin stress fibers became highly organized and aligned with the flow direction (**Figure 5A and B**). Notably, this rapid adaptation was quantitatively confirmed by the significant reduction in the interquartile range (IQR) of fiber orientation when increasing shear stress levels, regardless of the duration (2 hr or 24 hr) of flow exposure (**Figure 5—figure supplement 1A**).

Despite the flow-mediated alignment of the actin fibers, bacterial microcolonies adhering at the surface of endothelial cells in the VoC were able to induce the formation of the honeycomb-shaped actin structures underneath bacterial aggregates, similar to observations made in infection in the absence of flow (**Figure 5C and D**). This specific bacteria-induced reorganization of the actin cytoskeleton, which occurs under flow conditions, mirrors the flow-driven elongated shape of the bacterial microcolonies. After 3 hr of infection, 65% of the microcolonies had formed the actin cortical plaque below bacterial aggregates, while 35% did not show any visible structure (**Figure 5E**), which is close to the 2D experiments (**Figure 5E** and **Figure 5—figure supplement 1B**). The fluorescence intensity of F-actin beneath the microcolonies was twice as high compared to non-infected areas (**Figure 5F**), reaching levels previously observed in 2D models (**Soyer et al., 2014**). The highly organized actin fibers in the VoC were drastically reorganized by the presence of the bacterial microcolonies, as evidenced by the reduction in actin fiber coherency (fiber symmetry) upon infection (**Figure 5G**). Noticeably, F-actin fluorescence intensity was correlated with actin fiber reorganization (**Figure 5—figure supplement 1C**). Collectively, these results demonstrate that *N. meningitidis* reorganizes the endothelial cell actin cytoskeleton in the VoC under flow conditions, despite the presence of a pre-existing highly

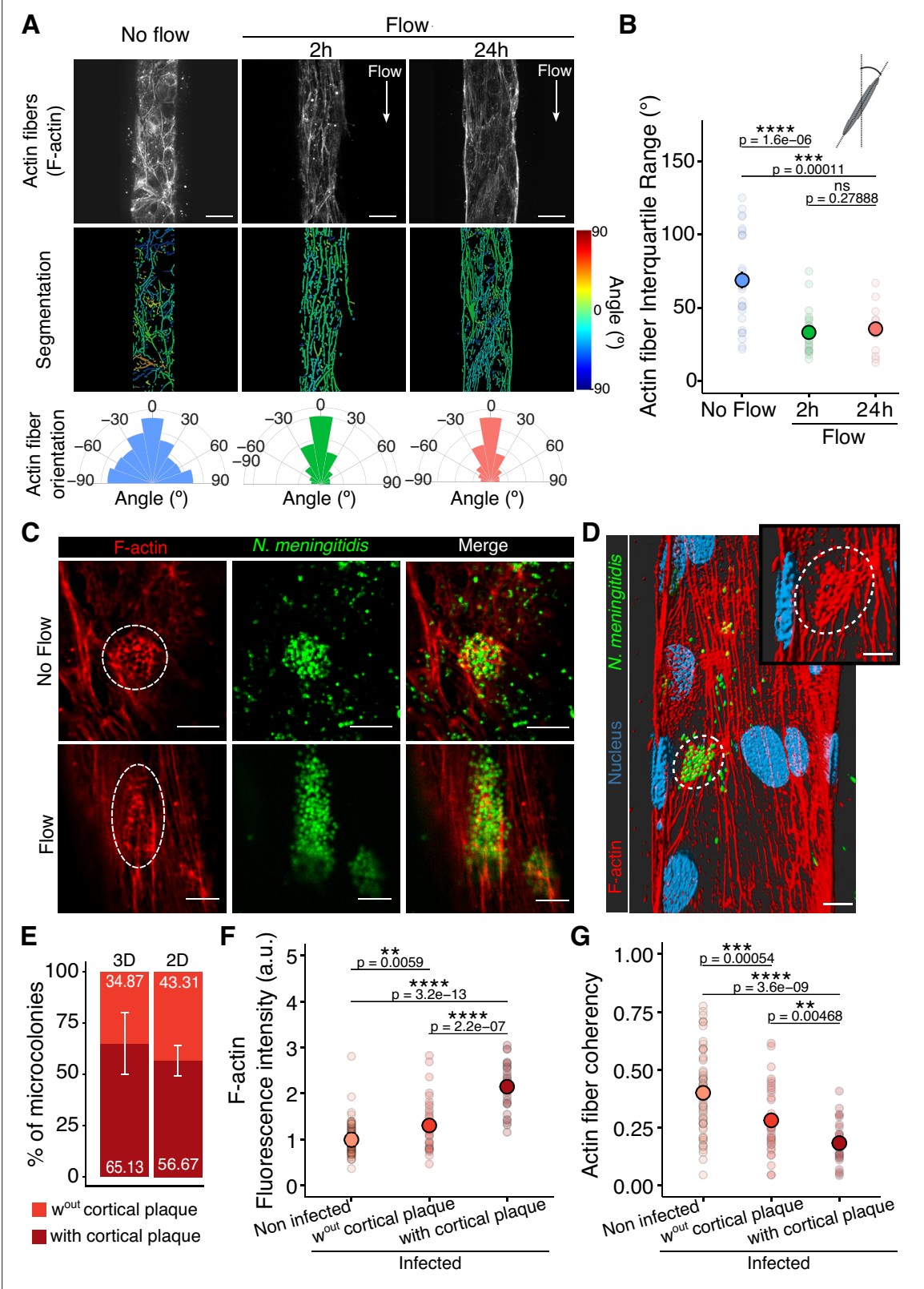

**Figure 5.** Flow-induced aligned actin stress fibers are reorganized below bacterial microcolonies. (**A**) Confocal images of the F-actin network in the Vessel-on-Chip (VoC) in the absence and presence of flow (2 hr and 24 hr) (top). Scale bar: 30 μm. Corresponding segmented images. The color code shows the alignment of the actin fibers with the direction of the flow (red to blue) (middle). Circular plots of the orientation distribution of actin fibers (bottom). (**B**) Interquartile range of F-actin fiber orientation. Each dot represents the mean of F-actin fiber orientation per vessel. For each condition, the

*Figure 5 continued on next page*

*Figure 5 continued*

mean ± s.d. is represented (No Flow: 68.8° ± 31.0° (n=29) — 2 hr of flow: 33.1° ± 14.3° (n=25) — 24 hr of flow: 35.5° ± 13.5° (n=19)). (**C**) Confocal images of honeycomb-shaped cortical plaque formed by *Nm* microcolonies in the absence (top) and presence of flow (bottom) in the VoC. Scale bar: 10 µm. (**D**) 3D rendering of a vessel infected with *Nm* 3 hr post-infection under flow conditions. Scale bar: 15 µm (main) and 10 µm (zoom). (**E**) Percentages of colonies forming a cortical plaque in the 3D VoC (n=5 vessels) and 2D regions of the chips (n=4 lateral channels). (**F**) F-actin fluorescence intensity under each microcolony and on non-infected regions of the same area. Each dot represents a region of a microcolony. For each condition, the mean ± s.d. is represented (Not infected regions: 0.99 ± 0.33 a.u. (n=74) — Infection, without cortical plaque: 1.30 ± 0.57 a.u. (n=36) — Infection, with cortical plaque: 2.15 ± 0.54 a.u. (n=28)). (**G**) Coherency of F-actin fibers on non-infected regions and infection sites. Each dot represents a colony. For each condition, the mean ± s.d. is represented (Not infected regions: 0.40 ± 0.17 a.u. (n=74) — Infection, without cortical plaque: 0.28 ± 0.15 a.u. (n=36) — Infection, with cortical plaque: 0.18 ± 0.09 a.u. (n=28)). All statistics have been computed with Wilcoxon tests.

The online version of this article includes the following figure supplement(s) for figure 5:

**Figure supplement 1.** Flow-induced aligned actin stress fibers are reorganized below *N. meningitidis* microcolonies.

organized and aligned F-actin network. This is of particular importance when considering the infection process in vivo – during which endothelial cells are submitted to flow-induced wall shear stress – leads to actin fiber alignment and high symmetry before bacterial adhesion.

## Neutrophils respond to *N. meningitidis* infection in a similar manner to that observed in the animal model

The presence of an inflammatory infiltrate, predominantly composed of neutrophils, in the vicinity of infected vessels is another cardinal feature of meningococcal infection (*Sotto et al., 1976*). Previous in vivo studies have demonstrated that *N. meningitidis* infection leads to the recruitment of neutrophils along infected venules, a process driven by the bacteria-mediated upregulation of E-selectin expression at the surface of infected endothelial cells (*Manriquez et al., 2021*). To assess whether this aspect of the infection can be recapitulated within the VoC upon meningococcal infection, we examined E-selectin expression in response to inflammatory stimuli and infection. Endothelial cells did not express E-selectin in the absence of infection (*Figure 6A*, *Figure 6—figure supplement 1A and B*), clearly demonstrating that the experimental setup successfully maintained appropriate basal conditions all along vessel formation. In contrast, both TNFα stimulation and *N. meningitidis* infection led to a statistically significant increase in E-selectin expression all over the vessel (*Figure 6A*, *Figure 6—figure supplement 1A and B*). This is due to both the increase in the number of E-selectin[+]-activated cells (*Figure 6B*) and the massive upregulation of E-selectin expression of these activated cells (*Figure 6C*) at the single-cell level. Interestingly, infection with the non-piliated *pilD* mutant strain induced E-selectin expression to a much lesser extent when compared to the wild-type strain, highlighting the role of bacterial pilus-mediated adhesion (*Figure 6A* and *Figure 6—figure supplement 1C*). The expression of E-selectin on *N. meningitidis*-infected endothelium was more spatially heterogeneous than on the TNFα-inflamed one, although there no direct spatial connection with bacterial adhesion (*Figure 6—figure supplement 1A and B*), suggesting a spatial cell-to-cell heterogeneity in endothelial cell response upon infection, which correlates with in vivo observations (*Manriquez et al., 2021*).

We next assessed the local recruitment of neutrophils to the infected on-chip model. To this end, human neutrophils were purified from the blood of healthy donors and perfused under flow conditions in the microfluidic chip (*Figure 6D*). *Figure 6E and F* show the increase in the numbers of neutrophils attaching to the VoC wall upon inflammatory stimulation, whether with TNFα or *N. meningitidis*-induced intravascular colonization, compared to control conditions, consistent with in vivo recruitment of neutrophils to infected vessels (*Melican et al., 2013*). We also observed phagocytosis of bacteria by human neutrophils (*Figure 6G*), which is comparable to that observed in the humanized mouse model (*Manriquez et al., 2021*). In addition, we could observe the typical cascade of human neutrophil adhesion events, including rolling, arrest, and crawling on the endothelial surface (*Figure 6—figure supplement 1D and E*), as previously described (*Ley et al., 2007*). Altogether, these results demonstrate that we have successfully recapitulated the full spectrum of human immune-endothelial cell interactions, from the adhesion cascade to neutrophil recruitment and phagocytosis at infection sites, demonstrating that our VoC is suitable for studying the immune response during *N. meningitidis* infection. Furthermore, in contrast to the humanized mouse model, which is limited by the interactions of murine immune cells with human-infected endothelial cells, our Vessel-on-Chip model has the

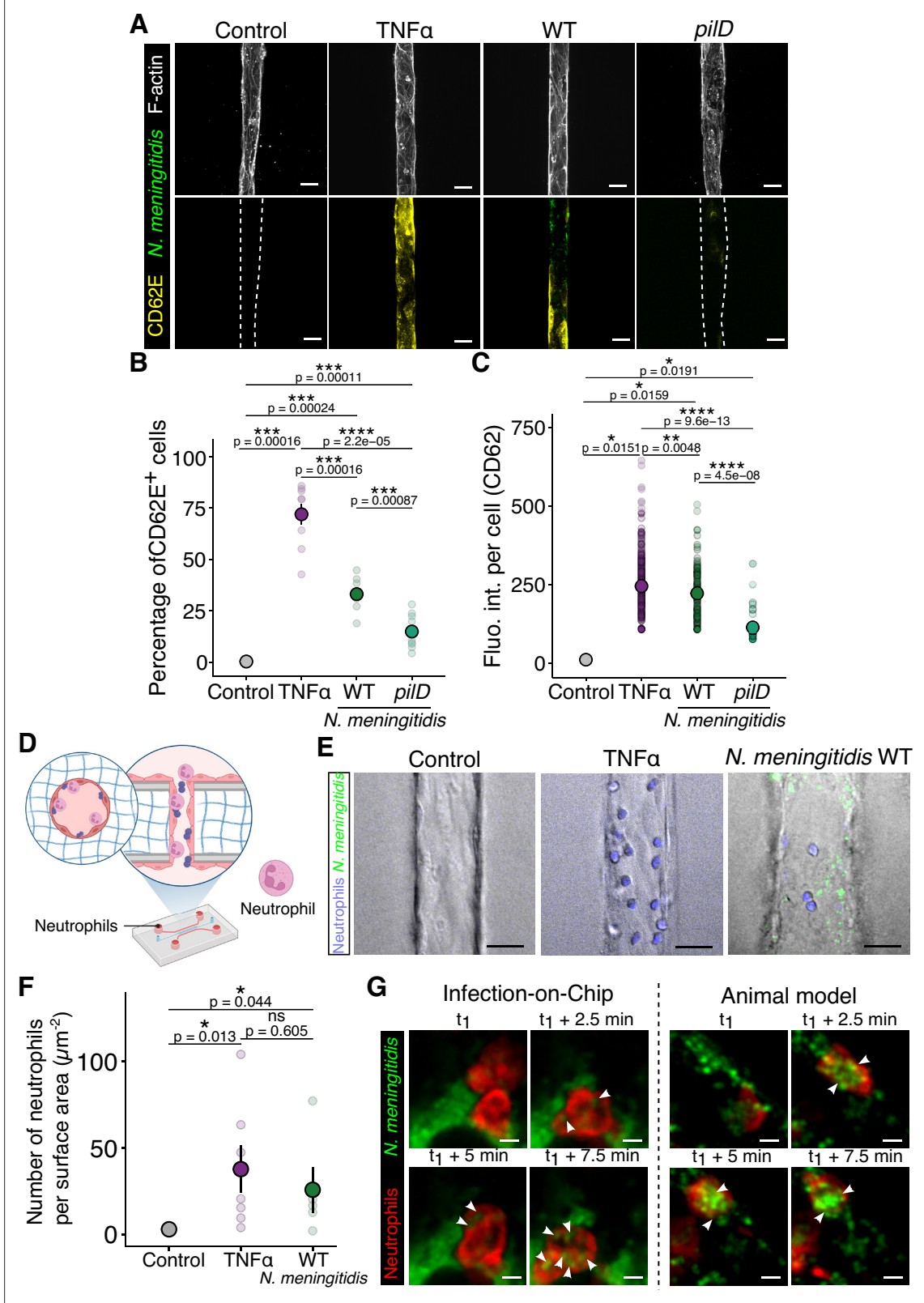

**Figure 6.** The infection model recapitulates the human neutrophil response to *N. meningitidis* infection. (**A**) Confocal images of E-selectin staining in the Vessel-on-Chip (VoC) for four conditions: without infection nor treatment, after 4 hr of either TNFα treatment or infection with wild-type (WT) or *pilD Nm* strains. Scale bar: 50 μm. (**B–C**) Graphs representing the percentage of E-selectin-positive cells and the mean intensity of CD62 in positive cells. For each condition, the mean ± s.d. is represented (Control: 0.32 ± 0.67%–(n=10 vessels), 11.4 ± 1.17 (n=2 cells) — TNFα: 72.0 ± 15.0%–(n=9

*Figure 6 continued on next page*

*Figure 6 continued*

vessels), 245 ± 100 (n=244 cells) — WT: 33.1 ± 8.12%–(n=8 vessels), 223 ± 118 (n=126 cells) — *pilD*: 14.9 ± 8.1%–(n=10 vessels), 144 ± 58.4 (n=44 cells)). (**D**) Schematic representation of the setup. Purified neutrophils were introduced in the microfluidic chip under flow conditions (0.7–1 μl/min). (**E**) Bright-field and fluorescence images of neutrophils adhering on a non-treated (control), TNFα-treated, or *Nm*-infected VoC. Scale bar: 25 μm. (**F**) Graph representing the number of neutrophils adhering to the endothelium for each condition. Each dot represents a vessel. For each condition, the mean ± s.d. is represented (Control: 0.59 ± 1.32 μm$^{-2}$ (n=5) — TNFα: 33.0 ± 36.1 μm$^{-2}$ (n=8) — *N. meningitidis*: 21.5 ± 28.4 μm$^{-2}$ (n=6)). (**G**) Representative images of bacteria phagocytosis by neutrophils in infected VoC (left) and in infected human vessels in the grafted mouse model (right). Scale bar: 25 μm. All statistics have been computed with Wilcoxon tests.

The online version of this article includes the following figure supplement(s) for figure 6:

**Figure supplement 1.** The infected Vessel-on-Chip model recapitulates the human neutrophil response to *N. meningitidis* infection.

potential to capture the human neutrophil response during vascular infections in a species-matched microenvironment.

## Discussion

In this work, we have developed an in vitro 3D human Vessel-on-Chip microfluidic-based platform that accurately mimics the vascular environment encountered by *N. meningitidis* during intravascular colonization. We not only characterized our model from the microfabrication to the cellular response under infection conditions, but we also demonstrated that our VoC strongly replicates key aspects of the in vivo human skin xenograft mouse model, the gold standard for studying meningococcal disease under physiological conditions. The *N. meningitidis*-infected VoC offers several distinct advantages described below, bridging the current gap between 2D in vitro and animal models for studying vascular infections.

The first advantage of our model is its ability to replicate complex in vivo-like vessel geometries. Although photoablation is often considered practically slow (*O'Connor et al., 2022*), our optimized process enables fast replication of complex geometries, thus facilitating the preparation of tens of devices in an hour. The speed capabilities drastically improve with the pulsing repetition rate. Given that our laser source emits pulses at 10 kHz, as compared to other photoablation lasers with repetitions around 100 Hz, our solution could potentially gain a factor of 100 in ablation speed. Also, we precisely controlled the size and shape of the UV-carved hydrogel scaffolds, enabling the replication of tissue-engineered VoC structures derived from intravital images, including straight, branched, and tortuous configurations. Vascular geometries, known to disturb streamlines and wall shear stress (*Davies, 2009*), have been implicated in the initial formation of atherosclerosis (*Zhou et al., 2023b*) and may also contribute to bacterial colonization in regions of low velocity (*Zhou et al., 2023a*). Our tunable system thus represents a unique tool to further explore how vascular geometries and shear stress impact vascular colonization in different conditions.

A second significant advantage of our VoC is the replication of endothelial tissue in an in vivo-like physiological state. This includes vessels of low permeability, with well-defined intercellular junctions, and able to dynamically maintain their integrity. Perfusing the microfluidic chips enables a flow-mediated wall shear stress, leading to the reorientation of actin fibers and the elongation of nuclei, similar to what we observed in the in vivo conditions. These findings demonstrate that our system successfully replicates the in vivo conditions of endothelial cells, emphasizing the physiological relevance of our approach and creating a realistic in vitro microenvironment to study *N. meningitidis* vascular infection.

In the Vessel-on-Chip, the colonization of *N. meningitidis* increases vessel permeability, also aligning with expectations from previous in vitro studies (*Kobsar et al., 2011*; *Endres et al., 2022*; *Schubert-Unkmeir et al., 2010*). In vivo, such an increase in permeability is only subtly visible after a few hours of infection but typically manifests at later stages (around 24 hr post-infection *Manriquez et al., 2021*). This difference is most likely associated with the presence of other cell types (e.g. fibroblasts, pericytes, perivascular macrophages) in the in vivo tissues and the onset of intravascular coagulation (*Corre et al., 2022*; *Obino and Duménil, 2019*). Coagulation has recently been successfully recapitulated in 3D vascular models under healthy conditions (*Middelkamp et al., 2023*), suggesting that our system could also be used to study the later stages of infection and the associated coagulation process.

Likewise, while perivascular cells and fibroblasts have been extensively studied in in vitro models (*Offeddu et al., 2019*; *Long et al., 2025*), their influence on *N. meningitidis*/host interactions have not been explored, and our setup would permit these types of studies. Recent studies show that substrate stiffness, through tissue remodeling, can affect bacterial adhesion (*Feng et al., 2023*) and uptake (*Liu et al., 2021*). We hypothesize that fibroblasts, by remodeling the ECM (*Zhao et al., 2024*), and perivascular cells, by enhancing mechanical stability and secreting the basement membrane (*Franca et al., 2025*), can alter vascular tissue and thus locally influence *N. meningitidis*-endothelium interactions, which can have larger consequences during vascular infection. Pericytes and endothelial cells can be co-seeded in the chips, as they have been shown to self-organize and spatially segregate within hours after seeding (*Long et al., 2025*). Fibroblasts, by contrast, must be embedded in the collagen before polymerization. Optimization is required to preserve chip integrity since fibroblasts generate contractile forces while migrating through the gel (*Lee et al., 2017*). Incorporating perivascular cells alongside other microvascular cell types within the physiological microenvironment of our model to dissect their roles in host–pathogen interactions will be of interest in the context of future studies.

The infected Vessel-on-Chip model also replicates the dynamics, morphology, and dependency of T4P for efficient intravascular colonization. *N. meningitidis* introduced in the VoC readily adheres along the endothelium and forms three-dimensional microcolonies that increase in size at a rate comparable to in vivo conditions. The microcolony shift in morphology we observed between no-flow and flow conditions is likely a consequence of the influence of shear stress on these visco-elastic structures (*Bonazzi et al., 2018*; *Welker et al., 2018*; *Craig et al., 2019*). While geometry-induced shear stress gradients did not strongly impact *N. meningitidis* adhesion, flow could have significant implications for the dynamics of vascular colonization by spreading adherent meningococcal microcolonies along the endothelium. Following their initial T4P-mediated adhesion at the surface of endothelial cells, meningococci subsequently reorganize the actin cytoskeleton within the on-chip model, leading to the formation of the typical honeycomb-shaped cortical plaque (*Merz and So, 1997*; *Soyer et al., 2014*). This sheds light on the capacity of *N. meningitidis* to reorient the initially organized, parallel actin network in the 3D microenvironment and underscore its importance for colony stabilization, as observed in 2D and in vivo (*Mikaty et al., 2009*; *Rossy et al., 2023*).

While this study focused on Type IV pili (T4P)-mediated adhesion, additional type IV pili-mediated functionalities, such as twitching motility can be explored using our VoC model. Twitching motility, driven by T4P retraction via the PilT ATPase, is crucial for bacterial colonization and microcolony formation on host surfaces (*Bonazzi et al., 2018*; *Eriksson et al., 2015*). The PilT mutant, which retains adhesion capabilities but lacks pilus retraction, serves as an ideal model to dissect the role of twitching motility in vitro. By comparing wild-type and PilT-deficient strains within the VoC, future studies could assess differences in bacterial movement, microcolony dynamics, and interactions with endothelial cells under shear stress conditions.

Finally, our model recapitulates the microenvironment required to study the human neutrophil response in contact with infected human endothelial cells. Unlike the humanized skin xenograft mouse model that involves heterotypic interactions between murine neutrophils and human endothelium (*Manriquez et al., 2021*), the VoC model allows for species-matched interactions between purified human neutrophils and human endothelial cells. While in basal conditions, no sign of inflammation was detected, *N. meningitidis* infection led to the upregulation of E-selectin on the endothelial surface and the increase of neutrophil adhesion on the vascular wall, leading to bacterial phagocytosis. These behaviors, including rolling, crawling, and active phagocytosis of adherent *N. meningitidis*, closely mirror immune responses observed in vivo (*Manriquez et al., 2021*). In addition, we observed that the *pilD* mutant strain, which lacks T4P, triggers a milder E-selectin upregulation, indicating that T4P-mediated adhesion plays a significant role in endothelial cell activation, thereby enhancing the subsequent neutrophil adhesion. The VoC model also facilitates quantitative measurements of key parameters, as illustrated by our ability to finely assess vessel permeability or detect E-selectin upregulation at the single-cell level, capturing both the number and intensity of individual activated cells. Reaching such a level of detail is often challenging in animal models due to geometric and resolution limitations; thus, our system represents a suitable alternative to facilitate both observations and quantification.

Beyond neutrophils, other immune cells, such as monocytes (*McNeil et al., 1994*), macrophages (*Joshi and Saroj, 2023*; *Varin et al., 2010*), and dendritic cells (*Villwock et al., 2008*) respond to *N. meningitis* and their interaction with meningococci can be revisited using our 3D model. For instance, macrophages are activated upon *N. meningitis* infection, eliminate the bacteria through phagocytosis, and produce pro-inflammatory cytokines and chemokines to activate other immune cells (*Joshi and Saroj, 2023*; *Varin et al., 2010*). Also, dendritic cells, uptaking *N. meningitidis*, activate adaptive responses through immune synapse formation with T cells (*Joshi and Saroj, 2023*), but meningococci were also shown to inhibit dendritic cell secretion of proinflammatory cytokines (e.g. TNF-α, IL-1β, IL-6, IL-8) (*Villwock et al., 2008*; *Rhodes et al., 2022*; *Jacobsen et al., 2016*). Such bacterial effects on immune responses, yet complex, would be interesting to study in our model, both because it offers a physiological 3D environment and because it potentially allows us to introduce different human immune cells, including B and T cells, absent in the skin-xenograft mice model.

To conclude, our Vessel-on-Chip system, which leverages 3D-engineered microenvironments, represents a significant advancement in studying bacterial physiology and pathology in realistic infection contexts. We have demonstrated that our model provides a 3D in vitro platform that recapitulates *N. meningitidis* vascular adhesion and microcolony formation within complex vascular morphologies, as well as the subsequent pathological consequences. Although the system was optimized for meningococcal infection, and given that the vasculature serves as a primary route for systemic infections (*Obino and Duménil, 2019*; *Rahman et al., 2022*) and bacterial dissemination to various organs (*Urbaniak et al., 2014*; *Nejman et al., 2020*), our approach paves the way for investigating the pathophysiological mechanisms triggered by other pathogens responsible for systemic infections (*Alonso-Roman et al., 2024*) (e.g. bacteria, viruses, fungi) in different vascular systems (e.g. capillary beds, arteries) (*Arakawa et al., 2020*). It is also suitable for both quantitative analysis and applied research in the biophysics and biomedical fields, including drug screening and antibiotic resistance studies. By enabling detailed exploration of previously overlooked aspects of infections (e.g. vascular geometry difference, cell specificity, virus/fungi/other bacteria), our model has the potential to advance the development of novel therapeutic strategies to better manage intravascular infections.

# Materials and methods
## Bacteria strains and culture

Infections were performed using *N. meningitidis* strains derived from the 8013 serogroup C strain (http://www.genoscope.cns.fr/agc/nemesys) (*Rusniok et al., 2009*). Mutations in the *pilD* gene have been previously described (*Rusniok et al., 2009*). Wild-type and *pilD* bacterial strains were genetically modified to constitutively express either the green fluorescent protein (GFP) (*Charles-Orszag et al., 2018*), the mScarlet fluorescent protein (mScar), or the near-infrared fluorescent protein (iRFP) under the control of the pilE gene promoter (*Bonazzi et al., 2018*).

Strains were streaked from –80°C freezer stock onto GCB agar plates and grown overnight (5% $CO_2$, 37°C). For all experiments, bacteria were transferred to liquid cultures in pre-warmed RPMI-1640 medium (Gibco) supplemented with 10% FBS at adjusted $OD_{600nm}$=0.05 and incubated with gentle agitation for 2 hr at 37°C in the presence of 5% $CO_2$.

*Escherichia coli* transformants were grown at 37°C on liquid or solid Luria-Bertani medium (Difco) containing 50 µg/ml kanamycin (for pCR-Blunt II-TOPO plasmid) or 20 µg/ml chloramphenicol plus 100 µg/ml ampicillin (for pMGC5-derived plasmid).

## pMGC24 construct

A plasmid allowing stable expression of the mScarlet-I fluorescent protein *Bindels et al., 2017* in *Neisseria meningitidis*, pMGC24, was constructed as follows: the sequence encoding the mScarlet-I fluorescent protein was PCR-amplified from the plasmid pmScarlet-I-C1 (Addgene #85044) with PacI and XhoI restriction sites in 5' and 3', respectively using the following primers: PacI_mScar_F: **TTAA TTAA***AGGAGTAATTTT*ATGGTGAGCAAGGGCGAG and XhoI_mScar_R: **CTCGAG**TTACTTGTACAG CTCGTCCATGC (with restriction sites **bolded** and RBS of *pilE* in *italic*). The PCR fragment was then cloned into the pCR-Blunt II-TOPO vector (Invitrogen). After sequencing of the insert, the plasmid was cut with PacI and XhoI, and the resulting fragment was ligated to PacI/XhoI-linearized pMGC5

plasmid (*Soyer et al., 2014*) that allows homologous recombination at an intergenic locus of the *Nm* chromosome and expression under the control of the constitutive *pilE* promoter.

## Chips production

The microfluidic chip was designed on Clewin 5.4 (WieWeb software). The corresponding photolithography mask was ordered from Micro Lithography Services LTD (Chelmsford, UK). The chip molds were created using photolithography on an MJB4 mask aligner (Süss MicroTec, Germany). Briefly, SUEX-K200 resin (DJ Microlaminates Inc, US) was laminated onto a 4-inch silicon wafer. Lamination temperature, baking, UV exposure dose, and development steps were performed according to the manufacturer's recommendations to produce molds with a 200 μm height. Obtained heights were verified with a contact profiler (DektakXT, Bruker). After overnight silanization (Trichloro(1 H,1H,2H,2H-perfluorooctyl) silane, Sigma-Aldrich), replication of the devices was performed with poly-dimethyl-siloxane (PDMS, Silgard 184; Dow Chemical). After polymerization, PDMS slabs were cut and punched to form inlets and outlets (1 and/or 4 mm² for lateral channels, 1 mm² for central channel). Bonding onto glass-bottom ibidi dishes (Ibidi, #81158) was performed with a Cute plasma cleaner (Femto science, Kr). Critically, assembled microfluidic chips were heated at 80°C for 48 hr to ensure both robust bonding and nominal PDMS hydrophobicity to allow consecutive hydrogel loading.

Chips were cooled down at 4°C for at least 8 hr to avoid quick polymerization while introducing the hydrogel in the chip. The collagen I solution was introduced in the center channel (see *Figure 1*). FujiFilm collagen I (LabChem Wako, Collagen-Gel Culturing Kit) has been used to obtain the 2.4 mg/ml solution, while the high concentration collagen I (Corning, #354249) has been used to obtain the 6 mg/ml, 4 mg/ml, and 3.5 mg/ml solutions according to the manufacturer's protocol. After collagen polymerization at 37°C for 20 min, PBS (Gibco) was added to the chips via the large lateral channels to avoid hydrogel drying. The inlet and outlet of the hydrogel channel were blocked with a droplet of liquid PDMS. Finally, chips were incubated at 37°C for 30 min to complete the hydrogel polymerization. After 30 min, PDB 1X is added in the chip to avoid gel drying.

## Photoablation

We designed the PDMS structures to allow further photoablation carving of 1–3 channels, maximizing the number of vessels that can be imaged while minimizing any loss of permeability at the PDMS/collagen/cells interface. The collagen I matrix was carved using our custom-made photoablation station. Briefly, the microfluidic chip is placed on the stage (SCANPLUS-IM, Marzhauser Wetzlar) of a double-deck microscope (IX81, Evident). A pulsed UV laser (MOPA-355, 500 mW, 10 kHz) is injected into the microscope through a custom-made side port using commercially available optics and opto-mechanical parts (Thorlabs). The laser is then focused within the gel bulk using a 20x objective with NA = 0.7 (UCPLFLN20X). The chip was then displaced with the stage at 1 mm/s to control the ablation region. Laser intensity was checked before each experiment using a light sensor (Thorlabs, Power Sensor Head S170C, and Power Meter for Laser, PM100A). Laser power under the sample was adjusted to 10 mW. The geometries of vessels have been designed with FUSION 360 (AutoCAD) or extracted from intravital microscopy images, exported as a .png binary image, and imported into a Python code. The control of the various elements is embedded and checked for this specific set of hardware. Notably, photoablation is a homemade fabrication technique that can be implemented in any lab equipped with an inverted microscope and a pulsed UV-laser with a repetition rate of around 10 kHz. While the optical setup might require some adjustment, the adaptation would be fairly standard.

Of note, Corning collagen gels above 6 mg/ml have not been assessed in VoC because they are too viscous and leak from the center channel, and those below 3.5 mg/ml are too liquid to be UV-carved (mechanically unstable). Also, too-high energy alters collagen by forming cavitation bubbles that degrade the scaffold edges, while a too-thin tube will avoid the entry of cells, both making it impossible to use.

## Chip endothelialisation and use

Primary human umbilical endothelial cells (HUVECs) were purchased from Lonza (pooled donors, #C2519A) and cultured in EGM-2 complete medium (Lonza, #CC-3162) in untreated 75 cm² flasks – 37°C, 5% CO₂. They were used between passages 2 and 6. Cells have been tested negative to

mycoplasma. Before passage or chip endothelialization, HUVECs were detached with 3 mL of Trypsin/EDTA (Gibco) for 4 min at RT. The cell/Trypsin/EDTA suspension was diluted with 4 mL of EGM-2 medium and centrifuged for 5 min at 1200 RPM (*i.e.*, 300 G). After supernatant removal, the cell pellet is diluted to 1/3 for passage and concentrated to $14 \times 10^6$ cells/mL for chip endothelialization.

Before endothelialization, chips were washed twice with a warm EGM-2 (with antibiotics) culture medium. The medium was partially removed (to avoid bubbles in the system) and 5 µl of the 14 M cells/mL suspension was introduced in the chip via one lateral channel (see *Figure 1*). Chips were then incubated for 30 min at 37°C and 5% $CO_2$. The same operation was repeated for the other lateral channel. After 30 more min of incubation, chips were gently washed with EGM-2 medium (with antibiotics) to remove non-adherent cells. Chips were kept at 37°C and 5% $CO_2$ for 12 hr.

The flow was added to the setup 12 hr after cell seeding: one inlet and one outlet were closed, syringes (SGE, 2.5 mL, glass) were connected to the inlet (needle: SGE Gastight Syringes, tubings: Tygon – Saint-Gobain, ID 0.02 mm and OD 0.06 mm/PTFE - Merck) and were controlled via a syringe-pump (Cetoni). Because the carved vessels are arranged in parallel (derivation) and the flow is controlled with a syringe pump (*i.e.,* controls directly the flow rate), the flow rate remains the same in each vessel regardless of the number of carved vessels per chip. Microfluidic chips were kept at 37°C - 5% $CO_2$ for at least 24 hr.

For TNFα treatment, a solution of 50 ng/mL (Sigma, #H8916) has been introduced in chips for 4 hr (37°C, 5% $CO_2$) to induce inflammation on the endothelium.

For bacterial infection, Vessel-on-Chips were rinsed three times with EGM-2 (without antibiotics) and kept at 37°C and 5% $CO_2$ overnight. After 2 hr of gentle agitation in EGM-2 (without antibiotics) culture medium as explained above, 10 µl of bacteria solution $OD_{600nm}$=0.05 is introduced in the Vessel-on-Chip. Flow was added to the setup following the description as above explained, during the 3 hr of infection, with a flow rate of 0.5 µl/min.

## Mice

SCID/Beige (CB17.Cg-PrkdcscidLystbg-J/Crl) mice were used in all the experiments (Central Animal Facility, Institut Pasteur, Paris, France). Mice were housed under the specific pathogen-free conditions at Institut Pasteur. Mice were kept under standard conditions (light 7 am - 7 pm; temperature 22 ± 1°C; humidity 50 ± 10%) and received sterilized rodent feed and water ad libitum. All experiments were performed in agreement with guidelines established by the French and European regulations for the care and use of laboratory animals and approved by the Institut Pasteur Committee on Animal Welfare (CETEA) under the protocol code CETEA 2018–0022. For all experiments, male and female mice between 6 and 13 weeks of age were used. Littermates were randomly assigned to experimental groups.

## Xenograft model of infection preparation

Five to eight weeks old mice, both males and females, were grafted with human skin as previously described (*Melican et al., 2013*). Briefly, a graft bed of about 1 cm² was prepared on the flank of anesthetized mice (intraperitoneal injection of ketamine and xylazine at 100 mg/kg and 8.5 mg/kg, respectively) by removing the mouse epithelium and the upper dermis layer. A 200-µm-thick human skin graft comprising the human epidermis and the papillary dermis was immediately placed over the graft bed. Grafts were fixed in place with surgical glue (Vetbond, 3 M, USA) and dressings were applied for 1 week. Grafted mice were used for experimentation 4–10 weeks post-surgery when the human dermal microvasculature is anastomosed to the mouse circulation without evidence of local inflammation, as previously described (*Melican et al., 2013*).

Normal human skin was obtained from adult patients (20–60 years old), both males and females, undergoing plastic surgery in the service de *Chirurgie Reconstructrice et Plastique* of Groupe Hospitalier Saint Joseph (Paris, France). Following the French legislation, patients were informed and did not refuse to participate in the study. All procedures were approved by the local ethical committee Comité d'Evaluation Ethique de l'INSERM IRB 00003888 FWA 00005881, Paris, France Opinion: 11048.

Then, intravital imaging of the human xenograft was adapted from *Ho et al., 2000*. Briefly, 30 min before surgery, mice were injected subcutaneously with buprenorphine (0.05 mg/kg) and anesthetized by spontaneous inhalation of isoflurane in 100% oxygen (induction: 4%; maintenance: 1.5% at 0.3 L/

min). A middle dorsal incision was made from the neck to the lower back, and the skin supporting the human xenograft was flipped and secured onto an aluminum custom-made heated deck at 36°C. The human microvasculature within the graft was exposed by carefully removing the excess connective tissue. The skin flap was covered with a coverslip and maintained thanks to a 3D-printed custom-made holder to avoid any pressure on the xenograft vasculature, sealed with vacuum grease, and continuously moistened with warmed 1× PBS (36°C). Mice's hydration was maintained by intraperitoneal injection of 200 µl 0.9% saline solution every hour. During the experiment, the mouse's body temperature was maintained at 37°C using a heating pad. The tail vein was cannulated, allowing the injection of fluorescent dyes and/or bacteria. In all experiments, human vessels were labeled using fluorescent conjugated UEA-1 lectin (100 µg, Dylight 650 or Dylight 755, Vector Laboratories), and five to ten fields of view of interest containing human vessels were selected per animal for observations.

### Ear mice preparation

Mice were anesthetized by spontaneous inhalation of isoflurane in 100% oxygen (induction: 4%; maintenance: 1.5% at 0.3 l/min) and placed in ventral decubitus on a heating pad (37°C) to maintain body temperature. The dorsal face of the ear pinnae was epilated using depilatory cream without causing irritation. The ear was carefully flattened out on an aluminum custom-made heated deck at 36°C and held in place with a coverslip. The tail vein was cannulated, allowing the injection of fluorescent dyes.

### Immunostaining

Collagen gels have been stained with a Maleimide Alexa Fluor 488 (Thermo Fisher, #A10254) solution: 10 µl of Maleimide (100 µg/ml) has been diluted within 700 µl of a 0.2 M sodium bicarbonate buffer solution. Chips were incubated with this solution for 1 hr at RT in the dark, and finally gently washed with PBS before imaging.

For E-selectin experiments, before and after TNFα treatment or bacterial infection, a solution of $10^{-3}$ mg/ml PE-conjugated anti-human E-selectin (CD62E clone P2H3, Thermo Fisher Scientific, #12-0627-42) has been introduced in the chips from the inlets and very little aspired from the outlets, and let for 10 min. After gentle washing steps with EGM-2 medium, E-selectin expression was acquired (Nikon, Spinning disk, Obj. 20X, dry, NA = 0.75).

Chips were fixed with PFA 4%v/v in PBS for 15 min, permeabilized with Triton X-100 1%v/v in PBS for 30 min, blocked with PBS-gelatin 1%v/v in PBS for 15 min, and mounted with fluoromount-G solution (Fluoromount-G, SouthernBiotech, #0100–01). Chips were incubated with Phalloidin at 0.66 µM (Alexa Fluor 488, Thermofisher, #A12379), anti-human PECAM antibody at 2.5 x $10^{-3}$ mg/ml (Ultra-LEAF Purified anti-human CD31 Antibody, Biolegend #303143 - conjugated with DyLight Antibody Labeling Kits, Thermo Fisher #53044), anti-human collagen IV antibody at $10^{-2}$ mg/ml (Alexa Fluor 642, Clone 1042, eBioscience #51-9871-82), rabbit anti-VE-cadherin (Abcam #ab33168) for 10 hr at 4°C. Alexa Fluor 555-conjugated donkey anti-rabbit antibody (Abcam #ab150074), and Hoechst $10^{-2}$ mg/ml (33362 trihydrochloride trihydrate, Invitrogen #H3570) have been added to the chips after three washing steps with PBS 1X, for 1 hr at room temperature.

Human and mouse skin samples were fixed with PFA 4% v/v in PBS overnight, washed three times in PBS 1X (30 min of incubation time for each side), and blocked with buffer (0.3% Triton + 1% BSA + 1% NGS in PBS) overnight at 4°C. Samples were stained for 3 days at 4°C with phalloidin (0.66 µM, Alexa Fluor 488, Thermo Fisher, #A12379) and overnight at 4°C with DAPI (0.3 mg/ml, Thermo Fisher, #62247). A sample clearing was made with a Rapid clear 1.47 medium for 2 days at room temperature, and a mounting solution was introduced between the slide and cover slide in rapid clear medium.

### Permeability assays

In VoC, 10 µl of a 0.1 mg/mL dextran solution (FITC, Sigma) has been introduced in chips for 30 min. Stacks of four images, spaced 1 µm apart, were acquired every minute.

In vivo, mice were intravenously injected with 20 µg of anti-mouse CD31 antibodies coupled with Alexa Fluor 647 (clone 390, Biolegend) to label blood vessels and the ear was allowed to stabilize for 30 min. Before imaging, 5 mg of fluorescein isothiocyanate (FITC) conjugated 70 kDa or 150 kDa dextran (Sigma-Aldrich, St Louis, MO) dissolved in 200 µL of phosphate-buffered saline (PBS) were injected intravenously. Stacks of 10 images, spaced 4 µm apart, were acquired every minute.

For both models, the permeability has been quantified following the equation:

$$P = \frac{R}{2} \frac{1}{\Delta I_0^{channel}} \frac{dI^{collagen}}{dt}$$

Where $R$ is the channel radius, $\Delta I_0^{channel}$ the first step in increase of fluorescence in the vessel, and $dI^{collagen}/dt$ the rate of increase in fluorescence outside the vessel.

Image analysis was conducted over the first 8 min following the initial fluorescence increase within the lumen for both in vivo and VoC models, while the entire acquisition lasted more than 30 min. For each condition, the mean fluorescence intensity was measured inside the vessel and in the surrounding area (background), both before and after introducing fluorescent Dextran. Due to reproducibility constraints in the animal model, only vessels without overlapping background vessels were included in the analysis.

## Imaging Particle Image Velocity and in vivo Wall Shear Stress

In vivo, basal microvascular blood flow in human vessels (arterioles and venules) was measured by high-speed acquisitions on a single plane (50 frames per second, 300 frames) of intravenously perfused (15 µl/min) 1 µm large fluorescent microspheres (Yellow/Green Fluoresbrite carboxylate, Polysciences, 107 microspheres/ml 1X PBS). Speed was determined from the centerline microsphere. The blood flow of each vessel was then computed from the vessel surface section and mean velocity following: $Q_{blood} = V_{bead}^{mean} \cdot S_{vessel}$.

According to *Figure 2—figure supplement 1B*, the flow rate $Q$ depends on vessel size $D$ as $1/D^3$ (the blue and red fitting curves are $1/x^3$). Therefore, we can compute the WSS ($\tau$) within circular structures following the *Equation 1*:

$$\tau = \frac{4\eta}{\pi} \frac{Q}{R^3} \tag{1}$$

where $R$ is the vessel radius, and the viscosity $\eta$ of blood vessel between 25 and 100 µm is considered as constant (*Pries et al., 1995*) ($\eta$=3.5 × 10$^{-3}$ Pa.s at 37°C). The WSS can then be averaged per vessel type (arterioles vs. venules) according to the equation (*Equation 1*). The same equation was used to compute the WSS in the Vessel-on-Chip from the flow rate $Q$.

## Neutrophil adhesion and phagocytosis

For experiments in the VoC, human neutrophils have been purified from donor blood. Human peripheral blood samples were collected from healthy volunteers through the ICAReB-Clin (Clinical Investigation platform) of the Institut Pasteur (*Esterre et al., 2020*). All participants received oral and written information about the research and gave written informed consent in the frame of the healthy volunteers CoSImmGEn cohort (Clinical trials NCT 03925272), after approval of the CPP Ile-de-France I Ethics Committee (2011, Jan 18th) Esterre2020. Neutrophils have been stained for nucleus (Hoechst, 10–3 mg/ml, 33362 trihydrochloride trihydrate, Invitrogen #H3570) and actin (SPY555-actin, 1/1000, Spirochrome #CY-SC202). After 10 min of a TNFα-induced activation (10 ng/mL), 10 µl of a 6.10$^6$ cells/mL suspension was introduced in the chips under flow. After the flow has been added to the system at 0.7–1 µl/min. Human neutrophil adhesion and phagocytosis have been acquired within 1 hr at 37°C and 5% $CO_2$.

In the animal model, after 2 hr of liquid culture, bacteria were washed twice in PBS and resuspended to 10$^8$ CFU ml$^{-1}$ in PBS. Before infection, mice were injected intraperitoneally with 8 mg of human transferrin (Sigma Aldrich) to promote bacterial growth in vivo, as previously described (*Rusniok et al., 2009*), and neutrophils were labeled by were labeled with 2.5 µg Dylight550-conjugated anti-mouse Ly-6G (LEAF purified anti-mouse Ly-6G, clone 1A8, Biolegend, #127620 with DyLight550 Antibody Labelling Kit, ThermoFisher Scientific, #84530). Mice were infected by intravenous injection of 100 µl of the bacterial inoculum (10$^7$ CFU total) and time-lapse z-stack series (2–2.5 µm z-step, 50–80 µm range) were captured every 20 min for 3 hr to image bacterial growth and every 30 s for 20 min to image phagocytosis of bacteria by neutrophils.

Straight channels have been mostly used for all experiments of the study. Rarely, we used the branched in vivo-like designs to observe potential similar infection patterns to in vivo, and related neutrophil activity, without noticing much difference.

## Rheology assays

Collagen gel viscoelastic properties were measured using an Ultra + rheometer (Kinexus). 200 µl of collagen gel was placed at the center of the rheometer plate kept at 4°C plate. The 20 mm-diameter testing geometry was lowered to a gap of 200 µm. Excess hydrogel was removed. Temperature was ramped up to 37°C (within 60 s) to initiate the hydrogel formation. Gelation dynamics and gel properties were extracted from continuous geometry rotation at 1 Hz frequency and a 1% strain. Rheological values i.e., elastic modulus G', stabilized after 200 s. Final values were determined by averaging values between 500 and 700 s.

## Actin fiber alignment/nucleus alignment and elongation

The z projection (Maximum Intensity) of the top and bottom parts of vessels has been realized for actin fibers and nucleus segmentation and analysis. Only the images taken with Leica Spinning disk microscope, 40X, NA = 1.10 (see below for more details) were used for analyses. The fibers and the nuclei located at the vessel edges have been excluded from the analysis. The angle and the ratio of major/minor axis have been measured for all vessels (circular plots) and averaged by vessel (dotted graph).

For in vivo images, the segmentation of nuclei has been achieved by hand for several z-slices for each vessel. The alignment (angle) and elongation (ratio major/minor) have been measured for all vessels (circular plots) and averaged by vessel (dotted graph).

## In vitro actin recruitment and coherency

The z projection (Maximum Intensity) of the top and bottom parts of the vessel (taken with the 40X, NA = 1.10 see below for more details) has been used to quantify actin recruitment (ac) and fibers coherency. Region of Interest (ROIs) of similar sizes were located on infected and non-infected areas of the endothelium. For each ROI, the mean fluorescence intensity has been subtracted from the background intensity (collagen matrix) and normalized by the mean of the non-infected areas' intensity:

$$I_{ac} = \frac{I_{ROI}^{infected} - I_{background}}{mean(I_{ROI}^{non-infected})}$$

The coherency has been assessed with the OrientationJ Measure plugin (BIG, EPFL).

## Simulations

The shear stress simulations have been performed with COMSOL Multiphysics 6.3. For each vessel, the contour has been segmented with Fiji ImageJ (ImageJ2, version 2.16.0/1.54 g, 26d66057dd), transformed into a binary image, and saved in a vector image (DXF) with Inkscape (Inkscape 1.4, e7c3feb1, 2024-10-09). The DXF file has been used to produce the 2D fluid simulation in COMSOL. Notably, the in vivo-like design must be rotated to allow the upper and lower branches of the complex structure to pass between the fixed PDMS pillars during photoablation. To remain consistent with the image and the flow direction, we have kept the same orientation as in the COMSOL simulation. This leads to a locally higher shear stress at the top of the architecture.

## Statistics

Post-processing and statistics analyses have been realized with R (R Studio version 2022.12.0+353, R version 4.2.2 (2022-10-31)). No statistical method was used to predetermine the sample size. Statistical tests were based on the Wilcoxon or t-tests – with $p$-values >0.05: n.s., 0.05<$p$-values<0.01: *, 0.01<$p$-values<0.005: **, 0.005<$p$-values<0.001: ***, $p$-values <0.001: ****. Scatter dot plots show the mean ± sd. Statistical details of experiments, such as sample size, replicate number, and statistical significance, are explicitly added in the figure legends and source data files.

## Acknowledgements

We thank P Vargas, A Desys, and M Bernard (Leukomotion, Institut Necker Enfants Malades) for their help in the purification of human neutrophils and C Travaillé (Photonic Bioimaging, Institut Pasteur) for complementary microfluidic chip imaging. We are grateful to the healthy volunteers of blood

donation, for their participation in the study. We thank ICAReB-Clin of the Medical Direction and ICAReB-biobank of the CRBIP (BioResource Center) of the Institut Pasteur (Paris) for providing blood samples from healthy volunteers. To H Laude, L Arowas, B L Perlaza, A Z M Delhaye and C Noury for managing the participants' visits. To E Roux, R Artus, L Sangari, D Cheval, and S Vacant for preparing the blood samples from donors. Funding was obtained by GD from the Fondation pour la Recherche Médicale (EQU202203014610), the DESTOP European Research Council (ERC) Advanced grant, the Fondation NRJ, and the 'Integrative Biology of Emerging Infectious Diseases' Labex (ANR 10-LBX-62 IBEID). Funding was obtained by GD and SG from the The Association Nationale pour la Recherche (ANR MeningoChip 18-CE15-0006-01). This work was also supported by DIM ELICIT's grant from Région Ile-de-France, obtained by SG.

## Additional information

### Funding

| Funder | Grant reference number | Author |
|---|---|---|
| Fondation pour la Recherche Médicale | EQU202203014610 | Guilllaume Dumenil |
| European Research Council | Destop | Guilllaume Dumenil |
| Fondation NRJ | Grand prix | Guilllaume Dumenil |
| Agence Nationale de la Recherche | ANR 10-LBX-62 IBEID | Guilllaume Dumenil |
| Agence Nationale de la Recherche | ANR MeningoChip 18-CE15-0006-01 | Guilllaume Dumenil |
| Région Ile-de-France | DIM Elicit | Samy Gobaa |

The funders had no role in study design, data collection and interpretation, or the decision to submit the work for publication.

### Author contributions

Léa Pinon, Conceptualization, Data curation, Formal analysis, Supervision, Validation, Investigation, Visualization, Methodology, Writing – original draft, Project administration, Writing – review and editing; Melanie Chabaud, Conceptualization, Data curation, Formal analysis, Investigation, Methodology; Pierre Nivoit, Conceptualization, Resources, Data curation, Formal analysis, Validation, Investigation, Visualization, Methodology; Jerome Wong Ng, Conceptualization, Resources, Software, Validation, Methodology, Writing – original draft, Writing – review and editing; Tri-Tho Nguyen, Resources, Data curation, Software, Formal analysis, Methodology; Vanessa Paul, Charlotte Bouquerel, Sylvie Goussard, Data curation, Formal analysis, Investigation, Methodology; Pauline Smilovici, Formal analysis, Investigation, Methodology; Emmanuel Frachon, Resources, Software, Methodology; Dorian Obino, Conceptualization, Data curation, Formal analysis, Funding acquisition, Validation, Investigation, Visualization, Methodology, Writing – original draft, Writing – review and editing; Samy Gobaa, Conceptualization, Resources, Software, Formal analysis, Supervision, Funding acquisition, Validation, Investigation, Methodology, Writing – original draft, Project administration, Writing – review and editing; Guilllaume Dumenil, Conceptualization, Resources, Formal analysis, Supervision, Funding acquisition, Validation, Investigation, Visualization, Writing – original draft, Project administration, Writing – review and editing

### Author ORCIDs

Léa Pinon ⓘ https://orcid.org/0000-0002-8645-071X
Melanie Chabaud ⓘ https://orcid.org/0000-0003-3167-7076
Pierre Nivoit ⓘ https://orcid.org/0000-0002-4893-6348
Jerome Wong Ng ⓘ https://orcid.org/0000-0001-8287-5203
Tri-Tho Nguyen ⓘ https://orcid.org/0000-0001-9991-7094
Dorian Obino ⓘ https://orcid.org/0000-0002-4108-2714

Samy Gobaa [ORCID] https://orcid.org/0000-0002-0125-8674
Guilllaume Dumenil [ORCID] https://orcid.org/0000-0001-9174-9110

### Ethics

All procedures were approved by the local ethical committee Comité d'Evaluation Ethique de l'IN-SERM IRB 00003888 FWA 00005881, Paris, France Opinion: 11048.

Reviewer #1 (Public review): https://doi.org/10.7554/eLife.107813.3.sa1
Reviewer #2 (Public review): https://doi.org/10.7554/eLife.107813.3.sa2
Reviewer #3 (Public review): https://doi.org/10.7554/eLife.107813.3.sa3
Author response https://doi.org/10.7554/eLife.107813.3.sa4

## Additional files

### Supplementary files

MDAR checklist

### Data availability

Data and Python codes are available on gitlab. (https://gitlab.pasteur.fr/wong/photoablationvessel2025, copy archived at *Wong, 2025*).

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
