## [Editor Report · eLife Assessment]

The authors develop an **important** microfluidic microvascular model called "Vessel-on-Chip", which they use to study *Neisseria meningitidis* interactions within this in vitro vascular system. **Compelling** evidence shows that the fabricated channels are lined by endothelial cells, and these can be colonized by N. meningitidis that in turn triggers neutrophil recruitment. This model has advantages over the human skin xenograft mouse model, which requires complex surgical techniques, however, it also carries limitations in that only endothelial cells and supplied specific immune cells in the microfluidics are present, while true vasculature contains a number of other cell types including smooth muscle cells, pericytes, and components of the immune system.

[Editors' note: this paper was reviewed by Review Commons.]

---

## [Referee Report · Reviewer #1 (Public review)]

Summary:

The work by Pinon et al describes the generation of a microvascular model to study *Neisseria meningitidis* interactions with blood vessels. The model uses a novel and relatively high throughput fabrication method that allows full control over the geometry of the vessels. The model is well characterized from the vascular standpoint and shows improvements when exposed to flow. The authors show that Neisseria binds to the 3D model in a similar geometry that in the animal xenograft model, induces an increase in permeability short after bacterial perfusion, and endothelial cytoskeleton rearrangements including a honeycomb actin structure. Finally, the authors show neutrophil recruitment to bacterial microcolonies and phagocytosis of Neisseria.

Strengths:

The article is overall well written, and it is a great advancement in the bioengineering and sepsis infection field. The authors achieved their aim at establishing a good model for Neisseria vascular pathogenesis and the results support the conclusions. I support the publication of the manuscript. I include below some clarifications that I consider would be good for readers.

One of the most novel things of the manuscript is the use of a relatively quick photoablation system. Could this technique be applied in other laboratories? While the revised manuscript includes more technical details as requested, the description remains difficult to follow for readers from a biology background. I recommend revising this section to improve clarity and accessibility for a broader scientific audience.

The authors suggest that in the animal model, early 3h infection with Neisseria do not show increase in vascular permeability, contrary to their findings in the 3D in vitro model. However, they show a non-significant increase in permeability of 70 KDa Dextran in the animal xenograft early infection. As a bioengineer this seems to point that if the experiment would have been done with a lower molecular weight tracer, significant increases in permeability could have been detected. I would suggest to do this experiment that could capture early events in vascular disruption.

One of the great advantages of the system is the possibility of visualizing infection-related events at high resolution. The authors show the formation of actin of a honeycomb structure beneath the bacterial microcolonies. This only occurred in 65% of the microcolonies. Is this result similar to in vitro 2D endothelial cultures in static and under flow? Also, the group has shown in the past positive staining of other cytoskeletal proteins, such as ezrin in the ERM complex. Does this also occur in the 3D system?

Significance:

The manuscript is comprehensive, complete and represents the first bioengineered model of sepsis. One of the major strengths is the carful characterization and benchmarking against the animal xenograft model. Beyond the technical achievement, the manuscript is also highly quantitative and includes advanced image analysis that could benefit many scientists. The authors show a quick photoablation method that would be useful for the bioengineering community and improved the state-of-the-art providing a new experimental model for sepsis.

My expertise is on infection bioengineered models.

Comments on revised version:

The authors have addressed all my concerns.

---

## [Referee Report · Reviewer #2 (Public review)]

Pinon and colleagues have developed a Vessel-on-Chip model showcasing geometrical and physical properties similar to the murine vessels used in the study of systemic infections. The authors succeed on their aim of developing an complex, humanized, in vitro model that can faithfully recapitulate the hallmarks of systemic infections.

The vessel was created via highly controllable laser photoablation in a collagen matrix, subsequent seeding of human endothelial cells, and flow perfusion to induce mechanical cues. This model could be infected with *Neisseria meningitidis* as a model of systemic infection. In this model, microcolony formation and dynamics, and effects on the host were very similar to those described for the human skin xenograft mouse model (the current gold standard for systemic studies) and were consistent with observations made in patients. The model could also recapitulate the neutrophil response upon N. meningitidis systemic infection.

The claims and the conclusions are supported by the data, the methods are properly presented, and the data is analyzed adequately. The most important strength of this manuscript is the technology developed to build this model, which is impressive and very innovative. The Vessel-on-Chip can be tuned to acquire complex shapes and, according to the authors, the process has been optimized to produce models very quickly. This is a great advancement compared with the technologies used to produce other equivalent models. This model proves to be equivalent to the most advanced model used to date (skin xenograft mouse model). The human skin xenograft mouse model requires complex surgical techniques and has the practical and ethical limitations associated with the use of animals. However, the Vessel-on-chip model is free of ethical concerns, can be produced quickly, and allows to precisely tune the vessel's geometry and to perform higher resolution microscopy. Both models were comparable in terms of the hallmarks defining the disease, suggesting that the presented model can be an effective replacement of the animal use in this area. In addition, the Vessel-on-Chip allows to perform microscopy with higher resolution and ease, which can in turn allow more complex and precise image-based analysis. The authors leverage the image-based analysis to obtain further insights into the infection, highlighting the capabilities of the model in this aspect.

A limitation of this model is that it lacks the multicellularity that characterizes other similar models, which could be useful to research disease more extensively. However, the authors discuss the possibilities of adding other cells to the model, for example, fibroblasts. The methodology would allow for integrating many different types of cells into the model, which would increase the scope of scientific questions that can be addressed. In addition, the technology presented in the current paper is also difficult to adapt for standard biology labs. The methodology is complex and requires specialized equipment and personnel, which might hinder its widespread utilization of this model by researchers in the field.

This manuscript will be of interest for a specialized audience focusing on the development of microphysiological models. The technology presented here can be of great interest to researchers whose main area of interest is the endothelium and the blood vessels, for example, researchers on the study of systemic infections, atherosclerosis, angiogenesis, etc. This manuscript can have great applications for a broad audience focusing on vasculature research. Due to the high degree of expertise required to produce these models, this paper can present an interesting opportunity to begin collaborations with researchers dealing with a wide range of diseases, including atherosclerosis, cancer (metastasis), and systemic infections of all kinds.

---

## [Referee Report · Reviewer #3 (Public review)]

Summary:

In this manuscript Pinon et al. describe the development of a 3D model of human vasculature within a microchip to study *Neisseria meningitidis* (Nm)- host interactions and validate it through its comparison to the current gold-standard model consisting of human skin engrafted onto a mouse. There is a pressing need for robust biomimetic models with which to study Nm-host interactions because Nm is a human-specific pathogen for which research has been primarily limited to simple 2D human cell culture assays. Their investigation relies primarily on data derived from microscopy and its quantitative analysis, which support the authors' goal of validating their Vessel-on-Chip (VOC) as a useful tool for studying vascular infections by Nm, and by extension, other pathogens associated with blood vessels.

Strengths:

• Introduces a novel human in vitro system that promotes control of experimental variables and permits greater quantitative analysis than previous models

• The VOC model is validated by direct comparison to the state-of-the-art human skin graft on mouse model

• The authors make significant efforts to quantify, model, and statistically analyze their data

• The laser ablation approach permits defining custom vascular architecture

• The VOC model permits the addition and/or alteration of cell types and microbes added to the model

• The VOC model permits the establishment of an endothelium developed by shear stress and active infusion of reagents into the system

Weaknesses:

• The VOC model contains one cell type, human umbilical cord vascular endothelial cells (HUVECs), while true vasculature contains a number of other cell types that associate with and affect the endothelium, such as smooth muscle cells, pericytes, and components of the immune system. However, adding such complexity may be a future goal of this VOC model.

Impact:

The VOC model presented by Pinon et al. is an exciting advancement in the set of tools available to study human pathogens interacting with the vasculature. This manuscript focuses on validating the model, and as such sets the foundation for impactful research in the future. Of particular value is the photoablation technique that permits the custom design of vascular architecture without the use of artificial scaffolding structures described in previously published works.

Comments on revised version:

The authors have nicely addressed my (and other reviewers') comments.

---

## [Author Response]

The following is the authors’ response to the original reviews

**Public Reviews:**

**Reviewer #1 (Public review):**
One of the most novel things of the manuscript is the use of a relatively quick photoablation system. Could this technique be applied in other laboratories? While the revised manuscript includes more technical details as requested, the description remains difficult to follow for readers from a biology background. I recommend revising this section to improve clarity and accessibility for a broader scientific audience.

As suggested, we have adapted the paragraph related to the photoablation technique in the Material & Method section, starting line 1147. We believe it is now easier to follow.

The authors suggest that in the animal model, early 3h infection with Neisseria do not show increase in vascular permeability, contrary to their findings in the 3D in vitro model. However, they show a non-significant increase in permeability of 70 KDa Dextran in the animal xenograft early infection. As a bioengineer this seems to point that if the experiment would have been done with a lower molecular weight tracer, significant increases in permeability could have been detected. I would suggest to do this experiment that could capture early events in vascular disruption.

Comparing permeability under healthy and infected conditions using Dextran smaller than 70 kDa is challenging. Previous research (1) has shown that molecules below 70 kDa already diffuse freely in healthy tissue. Given this high baseline diffusion, we believe that no significant difference would be observed before and after *N. meningitidis* infection, and these experiments were not carried out. As discussed in the manuscript, bacteria-induced permeability in mice occurs at later time points, 16h post-infection, as shown previously (2). As discussed in the manuscript, this difference between the xenograft model and the chip could reflect the absence of various cell types present in the tissue parenchyma or simply vessel maturation time.

One of the great advantages of the system is the possibility of visualizing infection-related events at high resolution. The authors show the formation of actin in a honeycomb structure beneath the bacterial microcolonies. This only occurred in 65% of the microcolonies. Is this result similar to in vitro 2D endothelial cultures in static and under flow? Also, the group has shown in the past positive staining of other cytoskeletal proteins, such as ezrin, in the ERM complex. Does this also occur in the 3D system?

We imaged monolayers of endothelial cells in the flat regions of the chip (the two lateral channels) using the same microscopy conditions (i.e., Obj. 40X N.A. 1.05) that have been used to detect honeycomb structures in the 3D vessels in vitro. We showed that more than 56% of infected cells present these honeycomb structures in 2D, which is 13% less than in 3D, and is not significant due to the distributions of both populations. Thus, we conclude that under both in vitro conditions, 2D and 3D, the amount of infected cells exhibiting cortical plaques is similar. These results are in Figure 4E and S4B.

We also performed staining of ezrin in the chip and imaged both the 3D and 2D regions. Although ezrin staining was visible in 3D (Author response image 1), it was not as obvious as other markers under these infected conditions, and we did not include it in the main text. Interpretation of this result is not straightforward, as the substrate of the cells is different, and it would require further studies on the behavior of ERM proteins in these different contexts.

**Author response image 1. sa4fig1:** F-actin (red) and ezrin (yellow) staining after 3h of infection with *N. meningitidis* (green) in 2D (top) and 3D (bottom) vessel-on-chip models.

**Recommendation to the authors:**

**Reviewer #1 (Recommendation to the authors):**
I appreciate that the authors addressed most of my comments, of special relevance are the change of the title and references to infection-on-chip. I think that the current choice of words better acknowledges the incipient but strong bioengineering infection community. I also appreciate the inclusion of a limitation paragraph that better frames the current work and proposes future advancements.The addition of more methodological details has improved the manuscript. Although as mentioned earlier the wording needs to be accessible for the biology community. I also appreciated the addition of the quantification of binding under the WSS gradient in the different geometries and shown in Fig 3H. However, the description of the figure and the legend is not clear. What does "vessel" mean on the graph and "normalized histograms ...(blue)" in the figure legend. Could the authors rephrase it?

In Figure 3F, we investigated whether *Neisseria meningitidis* exhibits preferential sites of infection. We hypothesized that, if bacteria preferentially adhered to specific regions, the local shear stress at these sites would differ from the overall distribution. To test this, we compared the shear stress at bacterial adhesion sites in the VoC (orange dots and curve) with the shear stress along the entire vascular edges (blue dots and curve). The high Spearman correlation indicates that there is no distinct shear stress value associated with bacterial adhesion. This suggests that bacteria can adhere across all regions, independently of local shear stress. To enhance clarity, the legend of Figure 3 and the related text have been rephrased in the revised manuscript (L289-314).

Line 415. Should reference to Fig S5B, not Fig 5B. Also, the titles in Supplementary Figure 4 and 5 are duplicated, and the description of the legend inf Fig S5 seems a bit off. A and B seem to be swapped.

Indeed, the reference to the right figure has been corrected. Also, the title of Figure S4 has been adapted to its contents, and the legend of Figure S5 has been corrected.

**Reviewer #2 (Recommendation to the authors):**
Minor comments to the authors:Line 163 "they formed" instead of "formed".Line 212 "two days" instead of "two day"Line 269 a space between two words is missing.

These three comments have been addressed in the revised manuscript.

In addition, I appreciate answering the comments, especially those requiring hypothesizing about including further cells. However, when discussing which other cells could be relevant for the model (lines 631 to 632) it would be beneficial to discuss not only the role of those cells but also how could they be included in the model. I think for the reader, inclusion of further cells could be seen as a challenge or limitation, and addressing these technical points in the discussion could be helpful.

We thank Reviewer #2 for the insightful suggestion. Indeed, the method of introducing cells into the VoC depends on their type. Fibroblasts and dendritic cells, which are resident tissue cells, should be embedded in the collagen gel before polymerization and UV carving. This requires careful optimization to preserve chip integrity, as these cells exert pulling forces while migrating within the collagen matrix. In contrast, T cells and macrophages should be introduced through the vessel lumen to mimic their circulation in vivo. Pericytes can be co-seeded with endothelial cells, as they have been shown to self-organize within a few hours post-seeding. These important informations are now included in the manuscript (L577-587).

**Reviewer #3 (Recommendation to the authors):**
Suggestions and RecommendationsSome suggestions related to the VOC itself:Figure 1, Fig S1, paragraph starting line 1071: More information would be helpful for the laser photoablation. For instance, is a non-standard UV laser needed? Which form of UV light is used? What is the frequency of laser pulsing? How many pulses/how long is needed to ablate the region of interest?

The photoablation process requires a focused UV-laser, with high frequency (10 kHz) to lower the carving time while providing the required intensity to degrade collagen gel. To carve a reproducible number of 30 µm-large vessels, we used a 2 µm-large laser beam at an energy of 10 mW and moved the stage (*i.e.*, sample) at a maximum speed of 1 mm/s. This information has been added to the related paragraph starting on line 1147 of the revised manuscript.

It is difficult to understand the geometry of the VOC. In Figure 1C, is the light coloration representing open space through which medium can flow, and the dark section the collagen? On a single chip, how many vessels are cut through the collagen? It looks as if at least two are cut in Figure 1C in the righthand photo.

In Figure 1C, the light coloration is the Factin staining. The horizontal upper and lower parts are the 2D lateral channels that also contain endothelial cells, and are connected to inlets and outlets, respectively. In the middle, two vertically carved 3D vessels are shown in the confocal image.

Technically, we designed the PDMS structures to allow carving of 1 to 3 channels, maximizing the number of vessels that can be imaged while minimizing any loss of permeability at the PDMS/collagen/cells interface. This information has been added in the revised manuscript (L. 1147).

If multiple vessels are cut in the center channel between the lateral channels, how do you ensure that medium flow is even between all vessels? A single chip with multiple different vessel architectures through the center channel would be expected to have different hydrostatic resistance with different architectures, thereby causing differences in flow rates in each vessel.

To ensure a consistent flow rate regardless of the number of carved vessels, we opted to control the flow rate directly across the chip with a syringe pump. During experiments, one inlet and one outlet were closed, and a syringe pump was used. Because the carved vessels are arranged in parallel (derivation), the flow rate remains the same in each vessel. If a pressure controller had been used instead, the flow would have been distributed evenly across the different channels. This has been added to the revised manuscript in the paragraph starting on line 1210.

The figures imply that the laser ablation can be performed at depth within the collagen gel, rather than just etching the surface. If this is the case, it should be stated explicitly. If not, this needs to be clarified.

One of the main advantages of the photoablation technique is carving the collagen gel in volume, and not only etching the surface. Thanks to the 3D UV degradation, we can form the 3D architecture surrounded by the bulk collagen. This has been added to the revised manuscript, lines 154-155.

Is the in-vivo-like vessel architecture connected to the lateral channel at an oblique angle, or is the image turned to fit the entire structure? (Figure 1F and 3E). Is that why there is high shear stress at its junction with the lateral channel depicted in Figure 3E?

All structures require connection to the lateral channels to ensure media circulation and nutrient supply. The in vivo-like design must be rotated to allow the upper and lower branches of the complex structure to pass between the fixed PDMS pillars. To remain consistent with the image and the flow direction, we have kept the same orientation as in the COMSOL simulation. This leads to a locally higher shear stress at the top of the architecture. This has been added in the revised manuscript, in the paragraph starting on line 1474.

Figure S1F,G: In the legend, shapes are circles, not squares. On the graphs, what do the numbers in parentheses mean?

Indeed, the terms "squares" have been replaced by "circles" in Figure 1. (1) and (2) refer to the providers of the collagen, FujiFilm and Corning, respectively. We have added this mention in the legend in Figure S1.

Figure 3B: how do the images on the left and right differ? Each of the 4 images needs to be explained.

The four images represent the infected VoC from different viewing angles, illustrating the three-dimensional spread of infection throughout the vessel. A more detailed description has been added in the legend of Figure 3.

Figure S3C is not referenced but should be, likely before sentence starting on line 299.

Indeed, the reference to Figure S3C has been added line 301 of the revised manuscript.

Results in Figure 3 with the pilD mutant are very interesting. It is worth commenting in the Discussion about how T4P functionality in addition to the presence of T4P contributes to Nm infection, and how in the future this could be probed with pilT mutants.

We thank Reviewer #3 for this relevant insight. Following adhesion, a key functionality of *Neisseria meningitidis* for colony formation and enhanced infection is twitching motility. As suggested, we have added in the Discussion the idea of using a PilT mutant, which can adhere but cannot retract its pili, in the VoC model to investigate the role of motility in colonization in vitro under flow conditions (L611–623).

Which vessel design was used for the data presented in Figures 4, 5, and 6 and associated supplemental figures?

Straight channels have been mostly used in figures 4, 5, and 6. Rarely, we used the branched in vivo-like designs to observe potential similar infection patterns to in vivo, and related neutrophil activity. This has been added in the revised manuscript, lines 1435-1439.

Figure 4B-D: the images presented in Figure 4C are not representative of the averages presented in Figures 4B,D. For instance, the aggregates appear much larger and more elongated in the animal model in Figure 4C, but the animal model and VOC have the colony doubling time (implying same size) in Figure 4B, and same average aggregate elongation in Figure 4D.

The images in Figure 4C were selected to illustrate the elongation of colonies quantified in Figure 4D. The elongation angles are consistent between both images and align with the channel orientation. Representative images of colony expansion over time, corresponding to Figure 4A and 4B, are provided in Figure S4A.

Figures 4E-F: dextran does not appear to diffuse in the VOC in response to histamine in these images, yet there is a significant increase in histamine-induced permeability in Figure 4F. Dotted lines should be used to indicate vessel walls for histamine, and/or a more representative image should be selected. A control set of images should also be included for comparison.

We thank Reviewer #3 for the insightful comment. We confirm that we have carefully selected representative images for the histamine condition and adjusted them to display the same range of gray levels. The apparent increase in permeability with histamine is explained by a slight rise in background fluorescence, combined with the smaller channel size shown in Figure 4E.

Figure S4 title is a duplicate of Figure S5 and is unrelated to the content of Figure S4. Suggest rewording to mention changes in permeability induced by Nm infection in the VOC and animal model.

Indeed, the title of Figure S4 did not correspond to its content. We have, thus, changed it in the revised manuscript.

Line 489 "...our Vessel-on-Chip model has the potential to fully capture the human neutrophil response during vascular infections, in a species-matched microenvironment", is an overstatement. As presented, the VOC model only contains endothelial cells and neutrophils. Many other cell types and structures can affect neutrophil activity. Thus, it is an overstatement to claim that the model can fully capture the human neutrophil response.

We agree with the Reviewer #3, that neutrophil activity is fully recapitulated with other cell types, such as platelets, pericytes, macrophages, dendritic cells, and fibroblasts, that secrete important molecules such as cytokines, chemokines, TNF-*α*, and histamine. In our simplified model we were able to reconstitute the complex interaction of neutrophils with endothelial cells and with bacteria. The text was modified accordingly.

Supplemental Figure 6 - Does CD62E staining overlap with sites of Nm attachment

E-selectin staining does not systematically colocalize with *Neisseria meningitidis* colonies although bacterial adhesion is required. Its overall induced expression is heterogeneous across the tissue and shows heterogeneity from cell to cell as seen in vivo.

Line 475, Figure 6E- Phagocytosis of Nm is described, but it is difficult to see. An arrow should be added to make this clear. Perhaps the reference should have been to Figure 6G? Consider changing the colors in Figure 6G away from red/green to be more color-blind friendly.

Indeed, the reference to the right figure is Figure 6G, where the phagocytosis event is zoomed in. We have changed it in the text. Adapting the color of this figure 6G would imply to also change all the color codes of the manuscript, as red has been used for actin and green for *Neisseria meningitidis*.

Lines 621-632 - This important discussion point should be reworked. Some suggested references to cite and discuss include PMID: 7913984, 15186399, 17991045, 18640287, 19880493.

We have introduced in the discussion parts the following references as suggested (3–7), and discussed more the importance of introducting of immune cells to study immune cell-bacteria interaction and related immune response (L659-678).

Minor corrections:• Line 8 - suggest "photoablation-generated" instead of "photoablation-based"• Line 57- remove the word "either", or modify the sentence• Sentence on lines 162-165 needs rewording• Lines 204-205- "loss of vascular permeability" should read "increase in vascular permeability"• Line 293- "Measured" shear stress, should be "computed", since it was not directly measured (according to the Materials & Methods)• Line 304- "consistently" should be "consistent"• Fig. 3 legend, second line: replace "our" with "the VoC"• Line 371, change "our" to "the"• Line 415- Figure 5B doesn’t appear to show 2-D data. Is this in Figure S5B? Some clarification is needed. The quantification of Nm vessel association in both the VOC and the animal model should be shown in Figure 5, for direct comparison.• Supplementary Figure 5C: correlation coefficient with statistical significance should be calculated.• Figure 6 title, rephrase to "The infected VOC model"• Line 450, replace "important" with "statistically significant"• Line 459, suggest rephrasing to "bacterial pilus-mediated adhesion"• Line 533- grammar needs correction• Line 589- should be "sheds"• Line 1106- should be "pellet"• Lines 1223-1224 - is the antibody solution introduced into the inlet of the VOC for staining? Please clarify.• Line 1295-unclear why Figure 2B is being referenced here

All the suggested minor corrections have been taken into account in the revised manuscript.

References

(1) Gyohei Egawa, Satoshi Nakamizo, Yohei Natsuaki, Hiromi Doi, Yoshiki Miyachi, and Kenji Kabashima. Intravital analysis of vascular permeability in mice using two-photon microscopy. Scientific Reports, 3(1):1932, Jun 2013. ISSN 2045-2322. doi: 10.1038/srep01932.

(2) Valeria Manriquez, Pierre Nivoit, Tomas Urbina, Hebert Echenique-Rivera, Keira Melican, Marie-Paule Fernandez-Gerlinger, Patricia Flamant, Taliah Schmitt, Patrick Bruneval, Dorian Obino, and Guillaume Duménil. Colonization of dermal arterioles by *Neisseria meningitidis* provides a safe haven from neutrophils. Nature Communications, 12(1):4547, Jul 2021. ISSN 2041-1723. doi: 10.1038/s41467-021-24797-z.

(3) Katherine A. Rhodes, Man Cheong Ma, María A. Rendón, and Magdalene So. Neisseria genes required for persistence identified via in vivo screening of a transposon mutant library. PLOS Pathogens, 18(5):1–30, 05 2022. doi: 10.1371/journal.ppat.1010497.

(4) Heli Uronen-Hansson, Liana Steeghs, Jennifer Allen, Garth L. J. Dixon, Mohamed Osman, Peter Van Der Ley, Simon Y. C. Wong, Robin Callard, and Nigel Klein. Human dendritic cell activation by *Neisseria meningitidis*: phagocytosis depends on expression of lipooligosaccharide (los) by the bacteria and is required for optimal cytokine production. Cellular Microbiology, 6(7):625–637, 2004. doi: https://doi.org/10.1111/j.1462-5822.2004.00387.x.

(5) M. C. Jacobsen, P. J. Dusart, K. Kotowicz, M. Bajaj-Elliott, S. L. Hart, N. J. Klein, and G. L. Dixon. A critical role for atf2 transcription factor in the regulation of e-selectin expression in response to non-endotoxin components of *Neisseria meningitidis*. Cellular Microbiology, 18(1):66–79, 2016. doi: https://doi.org/10.1111/cmi.12483.

(6) Andrea Villwock, Corinna Schmitt, Stephanie Schielke, Matthias Frosch, and Oliver Kurzai. Recognition via the class a scavenger receptor modulates cytokine secretion by human dendritic cells after contact with *Neisseria meningitidis*. Microbes and Infection, 10(10):1158–1165, 2008. ISSN 1286-4579. doi: https://doi.org/10.1016/j.micinf.2008.06.009.

(7) Audrey Varin, Subhankar Mukhopadhyay, Georges Herbein, and Siamon Gordon. Alternative activation of macrophages by il-4 impairs phagocytosis of pathogens but potentiates microbial-induced signalling and cytokine secretion. Blood, 115(2):353–362, Jan 2010. ISSN 0006-4971. doi: 10.1182/blood-2009-08-236711.